# Surveying the global landscape of post-transcriptional regulators

Kendra Reynaud[1], Anna M. McGeachy[2], David Noble[2], Zuriah A. Meacham[2] & Nicholas T. Ingolia [1,2] ✉

Numerous proteins regulate gene expression by modulating mRNA translation and decay. To uncover the full scope of these post-transcriptional regulators, we conducted an unbiased survey that quantifies regulatory activity across the budding yeast proteome and delineates the protein domains responsible for these effects. Our approach couples a tethered function assay with quantitative single-cell fluorescence measurements to analyze ~50,000 protein fragments and determine their effects on a tethered mRNA. We characterize hundreds of strong regulators, which are enriched for canonical and unconventional mRNA-binding proteins. Regulatory activity typically maps outside the RNA-binding domains themselves, highlighting a modular architecture that separates mRNA targeting from post-transcriptional regulation. Activity often aligns with intrinsically disordered regions that can interact with other proteins, even in core mRNA translation and degradation factors. Our results thus reveal networks of interacting proteins that control mRNA fate and illuminate the molecular basis for post-transcriptional gene regulation.

A network of proteins regulates the expression of messenger RNA (mRNA) to maintain homeostasis and adapt cell physiology to changing environments[1]. This network includes *cis*-acting mRNA sequence elements and *trans*-acting factors that bind the transcript to regulate its fate[2]. RNA-binding proteins (RBPs) determine whether an mRNA is translationally activated or repressed, localized to a specific region within the cell, or degraded[1]. RBPs can also remodel RNA structure and act as chaperones to prevent RNA aggregation[3,4]. Determining the effects of regulatory RBPs is critical to understanding post-transcriptional control of gene expression.

Efforts to identify RBPs and their mRNA targets have revealed general principles of protein-RNA interactions. Recurring RNA-binding domains (RBDs) can individually recognize four to nine nucleotide motifs in RNA, and often appear in combination to achieve greater specificity[5–7]. Protein-RNA crosslinking reveals a diverse mRNA interactome that includes many proteins without canonical RBDs[8], including ~700 high-confidence RNA-protein interactions in budding yeast[8,9]. Reciprocally, crosslinking and immunoprecipitation (CLIP) experiments have defined the mRNA targets for hundreds of these RBPs[2,10,11].

These approaches expose a dense web of interactions, suggesting complex patterns of post-transcriptional regulation, but not the functional impact of these proteins on their target mRNAs. Measuring how individual RBPs regulate their direct mRNA targets[12] and examining how these targets change when the protein is perturbed[13] do not provide a scalable approach to characterize the regulatory networks underlying post-transcriptional regulation.

The modular architecture of regulatory RBPs[5] has spurred the development of the tethered function assay to bypass the endogenous RNA specificity of RBPs and instead measure their activity on a heterologous reporter transcript[14]. This approach can interrogate the regulatory effects of RBPs, isolated domains, or cofactors that do not bind RNA directly[15]. In the tethering assay, candidate regulatory proteins are targeted to a reporter transcript using the specific, high-affinity interaction between a bacteriophage coat protein and a cognate RNA hairpin, obviating whatever interactions might recruit a protein to its endogenous targets. This independence from endogenous target mRNAs and compatibility with robust reporters makes the tethering assay well-suited to high-throughput characterization

[1]California Institute for Quantitative Biosciences, University of California, Berkeley, Berkeley, CA, USA. [2]Department of Molecular and Cell Biology, University of California, Berkeley, Berkeley, CA, USA. ✉e-mail: ingolia@berkeley.edu

of post-transcriptional regulators. Indeed, tethering assays have revealed over 50 regulatory proteins in a systematic analysis of 700 full-length human RBPs[16]. It is also amenable to unbiased screening, as demonstrated by the identification of almost 300 post-transcriptional regulators in trypanosomes[17].

In this Article we adapt the tethering assay to survey regulatory activity across the entire yeast proteome. Our approach allowed us to identify hundreds of proteins that modulate mRNA translation and stability, including highly active, non-canonical RBPs. We subdivided proteins and mapped their regulatory activity to particular domains and regions, in some cases uncovering effects that are not apparent in the context of the full-length protein. This fine resolution allowed us to identify protein domains and short peptide motifs enriched among the most active post-transcriptional regulators. Notably, although many active regulators were canonical RBPs, their regulatory activity generally mapped outside the RBD. Our systematic, functional characterization of post-transcriptional regulators in budding yeast expands our understanding of the complex network of proteins that control RNA metabolism.

## Results

### Functional analysis of post-transcriptional regulators

We set out to functionally assess the RNA regulatory activity of proteins across the entire yeast proteome through the tethered-function assay. Flow cytometry provides high-throughput, single-cell phenotypic measurements and enables large, pooled screens using fluorescence-activated cell sorting (FACS). FACS analysis of tethered-function assays relies on fluorescent protein reporters[17], and so we devised a budding yeast tethering assay coupled to a ratiometric fluorescence readout. We tethered a transcript encoding a yellow fluorescent protein (YFP) with five boxB hairpins in its 3′ untranslated region (UTR) to a candidate regulatory protein fused to the λN coat protein[18]. To control for non-specific changes in cell size and physiology, we normalized the YFP measurements against a red fluorescent protein (RFP) control expressed from a transcript that is not targeted by λN. Changes in the ratio of fluorescence intensity between the yellow reporter and the red control precisely measure specific regulatory activity affecting the targeted mRNA while controlling for global effects (Fig. 1a). To further control for the possibility that binding of λN itself affects the reporter, we normalized the fluorescence ratio of the tethered fusion constructs against a tethered HaloTag protein, which exhibits no inherent regulatory effect.

We validated our assay by measuring how well characterized regulators affected reporter expression. Tethered poly(A)-binding protein (Pab1 in budding yeast) enhances reporter expression by stabilizing mRNA[14] and promoting its translation[19]. We observed an approximately threefold target RNA activation by tethered Pab1-λN, relative to an inactive HaloTag-λN control. Conversely, the CCR4–NOT complex is responsible for the majority of cytosolic mRNA deadenylation[20], and tethering of the CAF1 deadenylase (Pop2 in budding yeast) greatly destabilizes target mRNAs[21]. We saw approximately fivefold reporter repression by tethered Pop2-λN (Fig. 1b,c). We further tested how the particular choice of the λN•boxB interaction pair affected our results by tethering Pab1 and Pop2 to reporters containing one PP7 hairpin using fusions with the PP7 coat protein (PP7cp)[18] (Fig. 1b and Extended Data Fig. 1a). Both PP7cp fusions showed similar activity on their cognate targets as λN, although Pop2-PP7cp repression appeared weaker than Pop2-λN repression, potentially due to the use of only a single PP7 hairpin (Fig. 1c). We went on to measure the activity of the RBP Sgn1, which is linked to translation by genetic interactions and co-immunoprecipitation with Pab1 (Extended Data Fig. 1b)[22]. We found that Sgn1 served as a powerful activator that upregulated YFP expression by over sixfold relative to RFP (Fig. 1d), in addition to modestly increasing RFP levels and cell size (Extended Data Fig. 1c–f). Sgn1 tethering increased YFP RNA abundance ~2.5-fold (Extended Data Fig. 1g); based on the larger

change we see in YFP fluorescence, we infer that it activates translation as well. These results confirm that our tethering assay provides robust and quantitative measurements of mRNA-specific regulatory activity, even in the face of additional non-specific effects on the cell, and thus provides a powerful tool for a high-throughput, proteome-wide survey of mRNA regulators.

### A proteome-wide survey of post-transcriptional regulators

We set out to comprehensively survey the yeast proteome for post-transcriptional regulators by creating a large pool of cells that each expressed one λN fusion construct, sorting these cells into subpopulations according to their fluorescence phenotypes, and quantifying the tethering constructs in each of these sorted groups by deep sequencing. Tethering protein fusions with regulatory activity would alter the fluorescence phenotype of the host cell, shifting it into a subpopulation with an unusually low or high fluorescence ratio (Fig. 1a), and altering its distribution across the sorted cells.

We began by generating a proteome-scale library of λN fusions that would enable unbiased discovery of regulatory proteins and identification of functional domains within these regulators. We reasoned that we could construct an unbiased λN fusion library directly from randomly fragmented genomic DNA as budding yeasts have a compact and intron-poor genome, and thereby obtain a uniform representation of all proteins. However, we required an additional selection for fragments matching the correct strand and frame of a gene. We generated fragments by transposon-mediated tagmentation[23,24] and selected fragments of ~500 base pairs to capture whole protein domains, which have a typical size of ~100 amino acids[25] (Fig. 2a and Extended Data Fig. 2a). We captured these fragments into a vector that required in-frame translation through the fragmented sequence to express a downstream selectable marker (Fig. 2b). We found that ten out of ten individual clones encoded in-frame fusions (Supplementary Data 1). We then transferred our fragment library into a λN fusion expression vector and added random, 25-nucleotide barcodes that identify each fragment uniquely (Supplementary Data 2)[26,27]. The mean fragment size in our barcoded λN fusion library was ~500 base pairs, consistent with the fragment size of the genomic DNA input (Fig. 2c), and contained at least one representative fragment from roughly half of all yeast genes.

We analyzed the regulatory activity of each individual protein fragment in our library by pooled transformation, flow sorting and sequencing. We separated a population of cells transformed with our λN fusion library into four subpopulations of equal size according to the YFP/RFP fluorescence ratio, isolated library plasmid DNA from sorted cells, and quantified the barcodes by next-generation sequencing (Fig. 2d). We expected activators to be enriched in bins with higher YFP/RFP ratios, while repressors should be enriched in bins with lower ratios.

Indeed, certain tethering constructs displayed a dramatic skew in their abundance across the sorted cells. For example, one fragment of the RBP Sbp1 was sorted almost entirely into the highest YFP gate, indicating that it strongly activated reporter expression (Fig. 2e). We saw a similar strong enrichment for fragments of Pab1 (Extended Data Fig. 2b), reproducing the positive effect of tethering full-length Pab1 (Fig. 1c). Conversely, fragments of the nonsense-mediated decay factor Ebs1 and the RNA destabilizing protein Cth1 acted as strong repressors that were found almost exclusively in the lowest YFP subpopulation (Fig. 2f and Extended Data Fig. 2c). To quantify this enrichment, we computed an 'activity score' for each fragment: a maximum likelihood estimate of its average fluorescence, expressed as a z-score relative to the overall population. These scores ranged from −1.9 for strong repressors like Ebs1 and Cth1 to +1.9 for strong activators like Sbp1 and Pab1. Most fragments in our library had activity scores close to zero, indicating little or no effect on reporter transcript expression (Fig. 2g and Supplementary Data 3). Activity scores were reproducible between two biological replicate screens (Extended Data Fig. 2d); fragments with adequate sequencing coverage (at least 1,000 total reads across

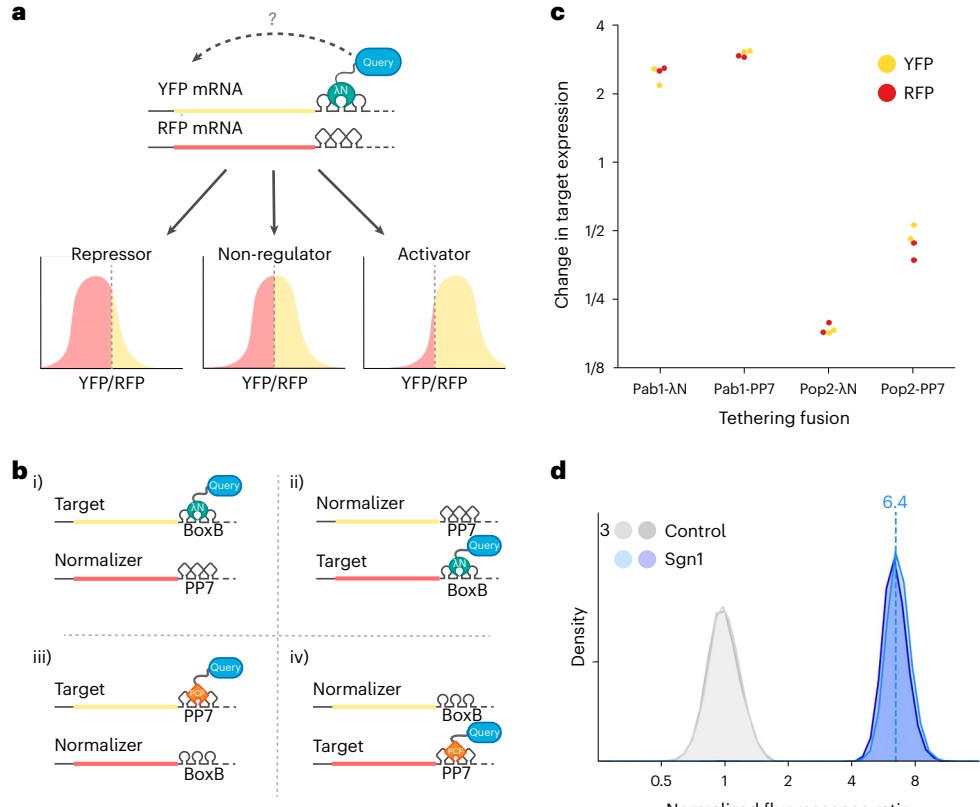

**Fig. 1 | The dual reporter tethering assay reports reproducible and quantitative regulatory effects. a**, Schematic of the tethering assay with a YFP reporter and RFP control (top), with expected fluorescence levels based on the activity of the tethered query protein (bottom). **b**, Schematic of testing the effects of the reporter mRNA and protein-RNA tether. **c**, The fluorescence ratio changes from tethering Pab1 and Pop2 to fluorescent protein reporter mRNAs, relative to tethering an inactive Halo control. Multiple regression (adjusted

$R^2 = 0.996$, $F(7, 8) = 563.2$, $P = 3.9 \times 10^{-10}$) indicated significant effects of Pab1 ($\log_2$ change of 1.35, 95% CI 1.18–1.53, $t = 17.8$, $P < 0.001$), Pop2 ($\log_2$ change of −2.40, 95% CI −2.58 to −2.23, $t = −31.7$, $P < 0.001$) and tether choice on Pop2 ($\log_2$ change of 0.92, 95% CI 0.57–1.26, $t = 6.0$, $P < 0.001$); no other terms were significant (all $P$ values are two-sided with no multiple testing correction). **d**, Distribution of fluorescence ratios reporting on the activity of Sgn1 in the tethering assay in two replicate samples. The dashed line represents the median YFP expression.

all bins) in both experiments had an activity score correlation of $r \approx 0.7$. We did note a linear rescaling of scores between the two screens, leading to saturation of strong activators and repressors in one replicate screen relative to the other. Because of this saturation effect, the strong correlation nonetheless underrepresents the actual agreement between the two screens. We relied on activity scores derived from the screen with broader dynamic range for our subsequent analysis.

We identified active fragments from many well-known regulatory proteins, such as the translation initiation factor Ded1[28–30] and Ngr1, which induces the decay of *POR1* mRNA[31]. Our unbiased approach also uncovered post-transcriptional regulation in proteins with other well characterized cellular functions, including the small heat shock chaperone Hsp26, which also has previously identified mRNA-binding activity[32]. Furthermore, we uncovered regulatory regions in proteins of unknown function, like Her1, which may interact with ribosomes based on co-purification experiments[33]. These results illustrate the power of our approach to discover proteins that control mRNA stability and translational efficiency and quantify how this affects gene expression.

**Full-length protein activity resembles truncated fragments**

We selected 12 fragments across a range of activity scores and biological functions (Fig. 3a) and directly measured their effect on reporter fluorescence. All 12 fragments shifted the fluorescence ratio in the direction expected from the large-scale survey (Fig. 3b), and the magnitude of the change correlated very well with their activity score ($r = 0.91$) (Fig. 3c and Extended Data Fig. 3a–e). This strong quantitative agreement

demonstrates that the activity score derived from sorting and sequencing is an accurate measure of the regulatory effect of a fragment.

Isolated protein fragments may have different activities than the full protein from which they are derived due to the absence of regulatory domains, altered protein-protein interactions, or other reasons. We thus selected a handful of active fragments to explore how fragment activity relates to the full protein. Sbp1 is an RBP with two RNA recognition motifs (RRMs) in addition to an arginine–glycine–glycine (RGG) motif that recruits Pab1[34]. The fragment that we characterized as an approximately threefold activator (Fig. 2e) contained only the first RRM and the RGG motif, whereas the full-length version of the protein was a weaker, approximately twofold activator (Fig. 3d). We hypothesize that the inclusion of the second RRM interferes with Pab1 recruitment, making it a weaker activator. In other cases, such as Sro9, the full-length protein had a stronger effect than the isolated fragment. Sro9 is an RBP that contains a La-motif and is hypothesized to activate translation through recruitment of the closed-loop-forming translation initiation complex[35]. We identified an Sro9 fragment that activated expression approximately twofold, whereas the full-length protein increased reporter expression by nearly fourfold (Fig. 3e). Tethering the entire yeast Puf-domain protein Jsn1 likewise produced a stronger repressive effect than the fragment we identified in our tethering library (Fig. 3f). In contrast, the intact version of the endocytic protein Yap1801[36] was less repressive than our fragment (Fig. 3g), perhaps because of differences in localization[37]. Nonetheless, in all four cases, the full-length protein exerted an effect in the same direction as

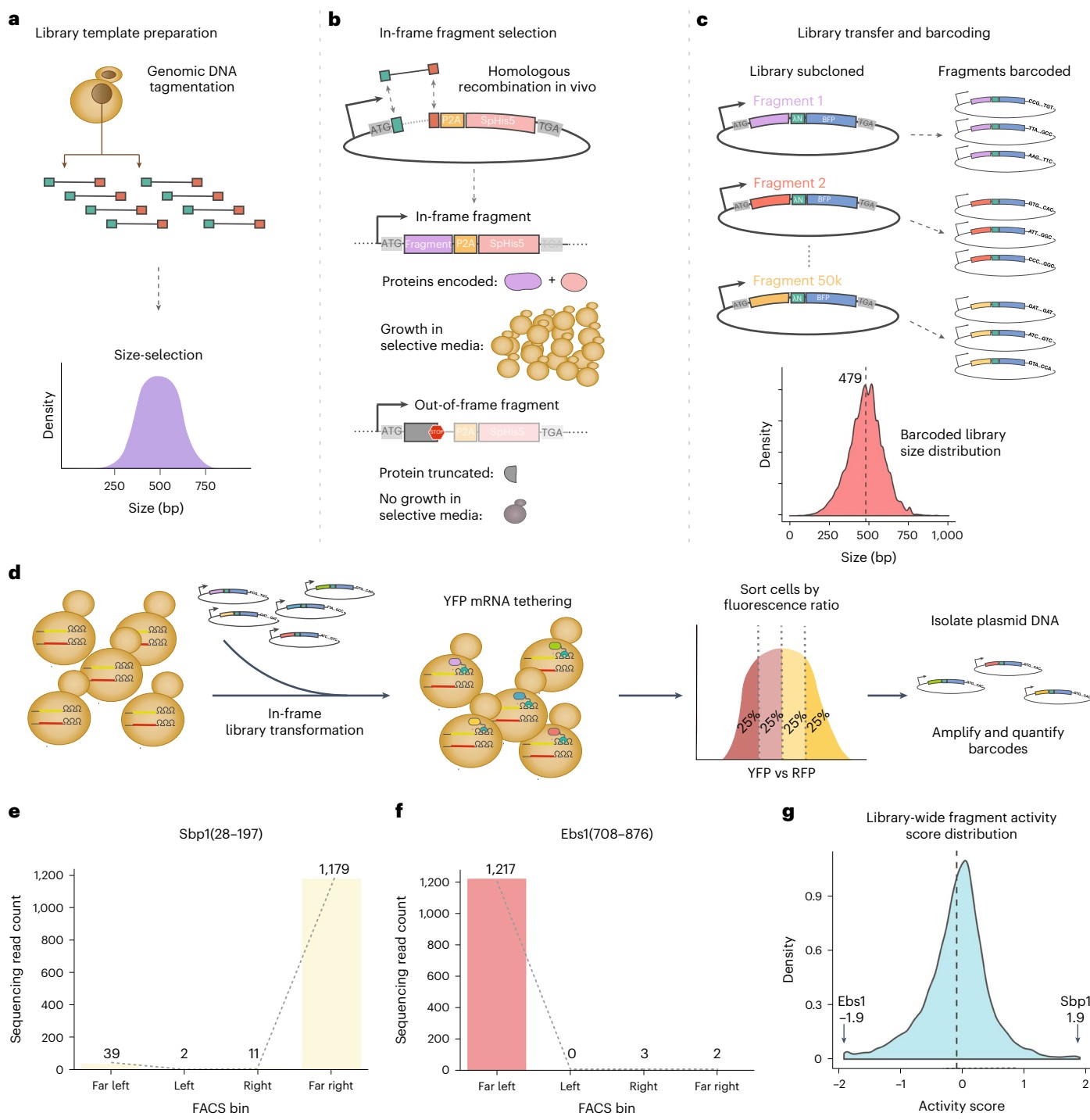

**Fig. 2 | Generating an unbiased, proteome-wide survey of tethered in-frame protein fragments. a**, S288C genomic DNA was fragmented by transposon-mediated tagmentation and selected to recover fragments with an average size of 500 base pairs. **b**, DNA fragments were cloned by in vivo gap repair into a plasmid containing a downstream selectable marker. Fragments containing an open reading frame in the correct phase will express a functional *Schizosaccharomyces pombe* HIS5 gene and support growth on selective media, whereas cells harboring out-of-frame fragments will fail to grow. **c**, Selected fragments were subcloned into the tethering vector with the λN and blue fluorescent protein (BFP) proteins encoded downstream. Fragments were assigned on average three barcodes each. The barcoded library size distribution did not change significantly from in the initial input fragment library. **d**, The library was transformed into the dual reporter yeast strain where the fragments were tethered to YFP mRNA. Phenotypic changes were captured by FACS sorting based on YFP versus RFP expression. Plasmids encoding the tethered fragments were isolated, and the barcodes associated with each fragment were amplified and then quantified through next-generation sequencing. **e**, Distribution of read counts per FACS bin for the Sbp1(28–197) activator fragment. **f**, As in **e**, for the Ebs1(708–876) repressor fragment. **g**, Kernel density estimate (KDE) of the library-wide activity score distribution.

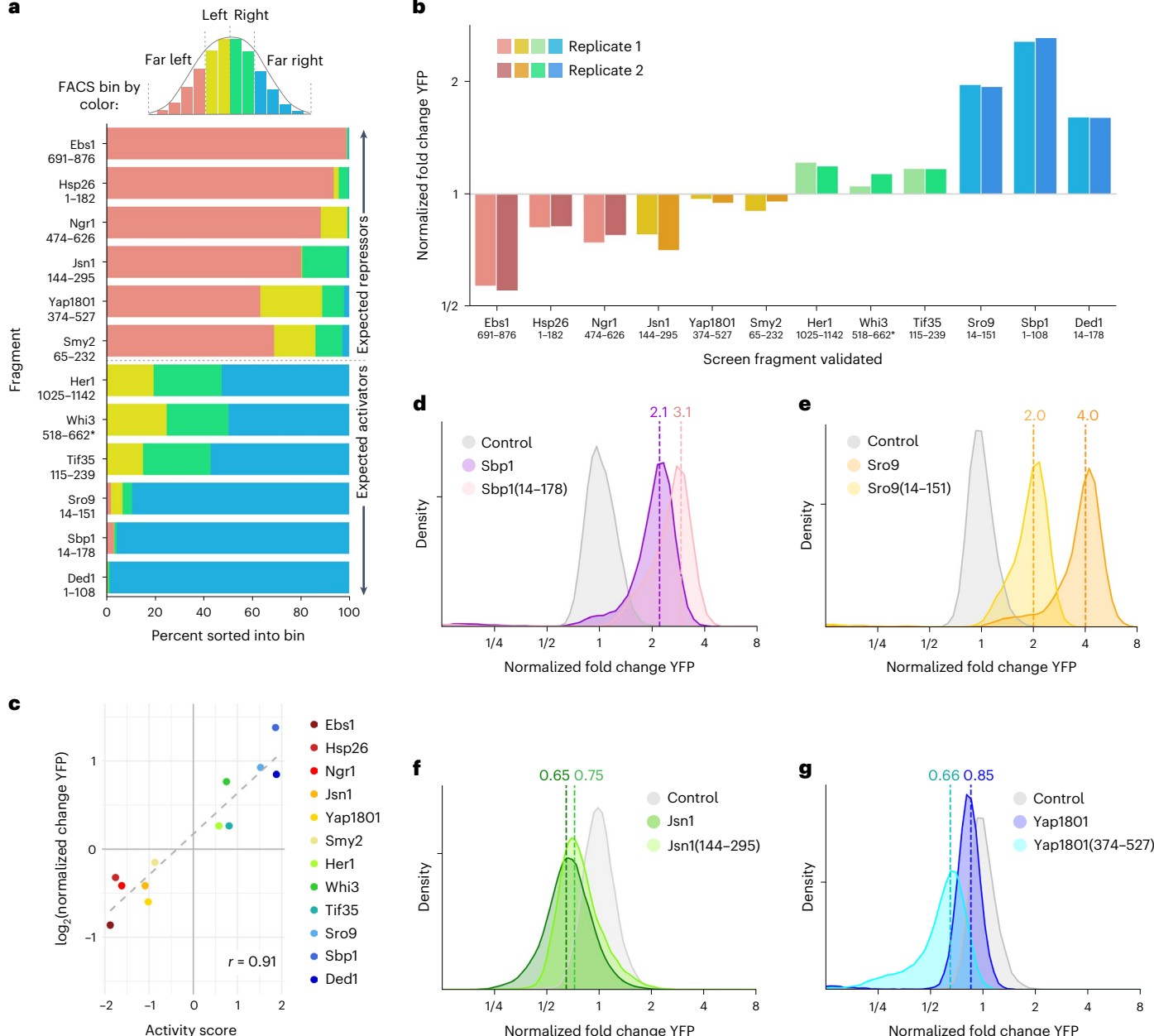

**Fig. 3 | Protein fragment activity in the tethering screen represents real, verifiable regulatory function. a**, Distribution of sequencing reads across subpopulations separated by FACS. **b**, Median activity of each protein fragment in the flow cytometry tethering assay ($n = 2$ per fragment). **c**, Comparison of the $\log_2$(difference in fluorescence ratio) and the screen activity score per fragment, $r = 0.91$. **d**, Flow cytometry measuring activity of Sbp1 and Sbp1(14–178) in the tethering assay ($n = 2$, one replicate per sample is shown). **e**, As in **d**, for Sro9 and Sro9(14–151) ($n = 2$, one replicate per sample is shown). **f**, As in **d**, for Jsn1 and Jsn1(144–295) ($n = 2$, one replicate per sample is shown). **g**, As in **d**, for Yap1801 and Yap1801(374–527) ($n = 2$, one replicate per sample is shown).

the fragment tested in our screen. Our approach is thus well suited to survey the regulatory activity contained in the native proteome and ascribe functions to RBPs.

## Activity in RBPs but not RBDs

Our tethering assay can detect regulatory activity in truncated proteins lacking RBDs and in co-regulator proteins that lack intrinsic RNA-binding activity. Nonetheless, we did expect a substantial overlap between the post-transcriptional regulators detected in the screen and known RBPs. To test this hypothesis, we compiled a list of budding yeast RBPs from proteins appearing in at least two of four RNA-protein interaction datasets (Fig. 4a)[9,38–41]. Fragments from these known RBPs

had substantially higher absolute activity scores than the overall proteome (Fig. 4b), further confirming the relevance of our results for endogenous programs of post-transcriptional regulation controlled by these RBPs. It also raised the question of whether regulatory activity was associated with the RBDs of these RBPs.

Our fragment library allowed us to ascribe quantitative regulatory effects to particular regions and domains within proteins. We were thus able to investigate which protein domains were enriched among the most active fragments in our screen, and whether these active regions coincided with RBDs. We identified fragments that contained at least 75% of some protein domain family from the Pfam database[42] and tested each family individually to determine whether the activity

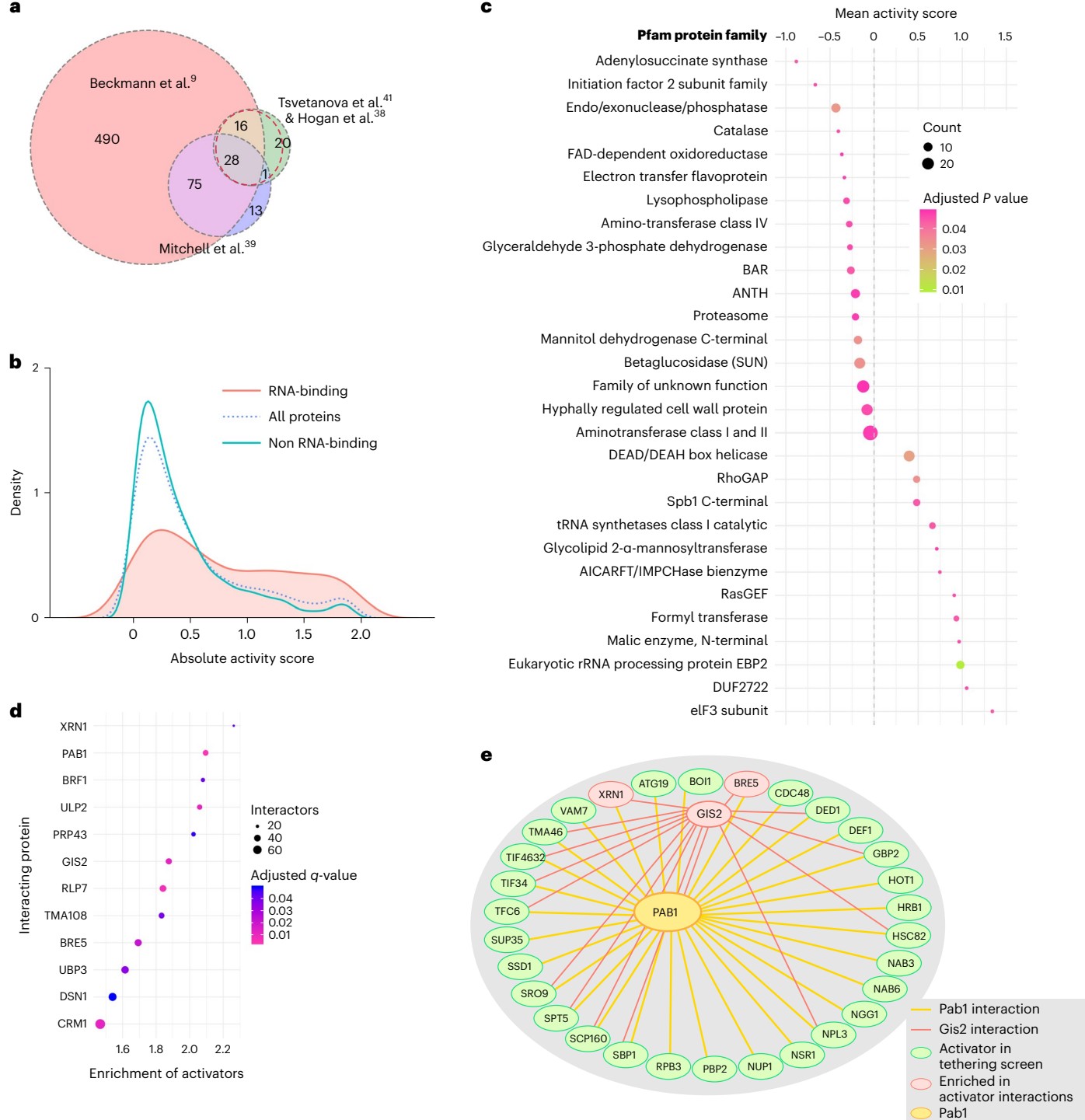

**Fig. 4 | Global analyses reveal enrichment of protein domains, motifs and protein–protein interactions amongst most active screen fragments. a**, Venn diagram of overlapping datasets identifying RBPs. **b**, KDE of absolute activity scores for RBPs, all proteins, and non-RBPs in the screen. **c**, Pfam protein domains significantly enriched amongst active screen fragments (adjusted *P* values from a two-sided Mann–Whitney U test with Benjamini–Hochberg multiple testing correction). **d**, Plot of proteins significantly enriched in interactions with activator fragments, where the *x* axis represents fold enrichment. **e**, Pab1 interactions with activator fragments (outer ring), with overlapping Gis2 interactions depicted in red.

scores of fragments containing that family were significantly higher or lower than the library overall (Fig. 4c).

Dozens of protein families were associated with active regulators, and some of the strongest associations involved domains with clear connections to translation and RNA decay (Supplementary Data 6). We observed the strongest positive mean activity score among fragments derived from the translation initiation factor eIF3[43]. We also saw a trend for activators among that DEAD box helicase family proteins, which include the translation initiation factors eIF4A and Ded1[44]. The endo/exonuclease/phosphatase family showed up among the strong repressors; these include certain subunits of the Ccr4–Not complex, for example[42]. We also saw many families encoding metabolic functions

such as adenylosuccinate synthetase[45], FAD-dependent oxidoreductase and the malic enzyme N-terminal domain. Metabolic enzymes have emerged as cryptic RBPs[9], and so it seems noteworthy that they appear to show regulatory activity as well. Notably, although many canonical RBDs such as RRMs appear in Pfam, they were not enriched in the active fragments. Canonical RNA-recognition domains appear more important for mRNA target selection, and regions outside of the RNA-interacting domains typically provide regulatory activity for RBPs.

Our screen also identified strong activity in fragments lacking an identifiable, folded domain. Indeed, many proteins contain intrinsically disordered regions (IDRs), which play important roles in post-transcriptional regulation[46]. In some cases, IDRs form protein–protein interactions, as in the case of the disordered N terminus of Ded1[47,48], whereas others serve as flexible linkers[49,50]. Functional IDRs can include short linear interaction motifs (SLiMs), which are often responsible for protein–protein interactions[51]. Although SLiMs are distinct from Pfam domains, they may be recognizable as peptide sequence motifs.

Motivated by the possibility that SLiMs could explain regulatory effects, we searched for peptide motifs enriched in active fragments using the MEME tool, and then scanned the yeast proteome for occurrences of these motifs using FIMO[52]. Some motifs were highly repetitive; although these repetitive motifs may have regulatory activity, it is difficult to interpret them, so they were excluded. We identified six non-degenerate motifs and repressors (Extended Data Fig. 4a,b and Supplementary Data 7 and 8), which align to genes with functions spanning many aspects of cell biology, including cell wall maintenance, cytoskeleton functions, transcription and translation. The glutamine-rich motif (repressor motif 2 in Extended Data Fig. 4a) is particularly enriched in genes involved in mRNA metabolism, such as *NGR1*, *POP2* and *PUF3*, which all have diverse roles in mRNA deadenylation and decay[31,53,54]. Likewise, the RGG repeat in activator motif 5 (Extended Data Fig. 4b) is widespread among RBPs and is linked to post-transcriptional regulation[55].

Regulatory RBPs often exert their effects by recruiting and activating core cellular machinery involved in translation and RNA decay. We thus expected that distinct active fragments from our screen might share common interactors. We intersected our library fragments with the physical protein–protein interactions in the BioGRID database[56] and searched for proteins with a significant over-representation of activating or repressing fragments among their interactors. We identified a dozen proteins enriched for interaction with activators (Supplementary Data 9), most tied clearly to RNA biology (Fig. 4d). Strikingly, the poly-(A) binding protein Pab1 showed one of the highest degrees of enrichment[19,57]. The translation regulator Gis2[58,59] was also substantially enriched in activators, and shared many interaction partners with Pab1 (Fig. 4e). Surprisingly, the exonuclease Xrn1 exhibited the strongest enrichment in activator interactions (Fig. 4d), despite its role in mRNA decay[60]. This enrichment may reflect a common core of mRNA-binding proteins that accompany transcripts during both translation and degradation. Alternatively, Xrn1 is reported to promote the translation of some transcripts encoding membrane proteins, and so this enrichment might also represent a more direct effect[61].

### Endoplasmic reticulum/Golgi protein Gta1 is a bimodal repressor

Several overlapping, C-terminal fragments of the protein Gta1 harboring a repressor-associated peptide emerged as potent repressors (Figs. 4b and 5a and Supplementary Data 5). Although the Gta1 protein co-purifies with the translational machinery[33], genetic evidence links it to golgi and vesicle transport[62,63], and the Gta1-GFP fusion protein localizes to the endoplasmic reticulum (ER)[64–66]. Owing to its reported association with ribosomes and the presence of a repressive motif, we generated λN fusion constructs of the strongly repressive

Gta1(603–767) fragment and the full-length Gta1 protein, and tested their effects on reporter expression (Fig. 5b).

Both full-length Gta1 and the Gta1(603–767) fragment robustly reduced median YFP and produced a strongly bimodal distribution of reporter expression (Fig. 5c and Extended Data Fig. 5a), a distinctive pattern that we did not see for any other tethering construct we examined individually. As expression of the isolated Gta1(603–767) fragment slowed cell growth, we focused our analysis on full-length Gta1. Gta1 tethering greatly reduced reporter mRNA abundance (Fig. 5d), suggesting that it promoted mRNA turnover. To track how bimodality emerged when the Gta1-λN fusion was switched on acutely, we expressed it from an inducible promoter. Levels of the YFP reporter began to decline within 1 h of inducing the tethering construct, and clear bimodality emerged within 2 h (Fig. 5e); continuing decline of reporter levels in the lower peak probably reflects the loss of pre-existing YFP through degradation or dilution. Notably, deletion of the repressor fragment that we identified in a Gta1Δ603–767-λN tethering construct abolishes this effect entirely (Fig. 5f,g), confirming that the Gta1(603–767) region containing our repressive peptide motif is both necessary and sufficient for its regulatory effect.

We next tested whether the bimodal reporter expression resulted from variation in the abundance of the Gta1 tethering fusion. Indeed, we saw a broad, bimodal distribution of blue fluorescent protein (BFP) fluorescence from the Gta1-BFP-λN construct after 4 h of induction (Fig. 5h), with levels increasing uniformly in the first hour of induction, followed by the emergence of two distinct phenotypes (Fig. 5h). Notably, we saw a similar trajectory after induction of the Gta1Δ603–767-λN tethering construct (Fig. 5i), although it did not affect YFP expression (Fig. 5f). We also measured the mRNA abundance of inducible *GTA1*, which quickly rose upon induction, then declined substantially after 4 h in the continuous presence of the inducer (Fig. 5j). In contrast, levels of the mRNA encoding the inactive Halo-BFP-λN tethering control increased steadily in the 2 h following induction (Extended Data Fig. 5b). These mRNA abundance measurements reflect population averages, whereas flow cytometry highlights the cell-to-cell variability.

We also noted that induction of full-length *GTA1*, or the *GTA1Δ603–767* mutant lacking RNA destabilization activity, cause an atypical, elongated morphology and persistent clusters of cells, akin to the filamentous growth that *Saccharomyces cerevisiae* can undergo upon starvation (Fig. 5k)[67,68]. Because Gta1Δ603–767 is not a strong repressor, but still impacts budding morphology (Fig. 5i), RNA turnover appears separable from budding effects.

### IDRs mediate regulatory activity

Our library contained many fragments of Ccr4, one of two deadenylases in the Ccr4–Not complex and thus a key mRNA decay factor[69]. Consistent with this role, we identified many repressive Ccr4-derived fragments; the median activity score of Ccr4 fragments was −0.5, and the strongest repressive fragment, Ccr4(2–203), had an activity score of −1.8 (Fig. 6a,b). This strongly repressive fragment originated from the N terminus of Ccr4 rather than the C-terminal nuclease domain[70,71]. Indeed, the disordered N terminus yielded the strongest repressors, while the adjoining, folded leucine-rich repeat had little activity on its own (Fig. 6c and Extended Data Fig. 6a). Our results suggest a regulatory role for the disordered N terminus, which is not required for Ccr4 nuclease activity or assembly into the Ccr4–Not complex[71].

A similar pattern emerged among the regulatory fragments derived from the translation initiation factor Ded1. This highly conserved RNA helicase of the DEAD-box family interacts with core translation initiation factors in the cap-binding eIF4F complex and is important for translation of many yeast mRNAs[47,48,72,73]. In agreement with its positive role in mRNA expression, Ded1 fragments appeared among the strongest of the post-transcriptional activators (Fig. 6d and Extended Data Fig. 6b). Longer fragments of the disordered N terminus of Ded1 had greater activity (Fig. 6e,f), consistent with deletion analyses

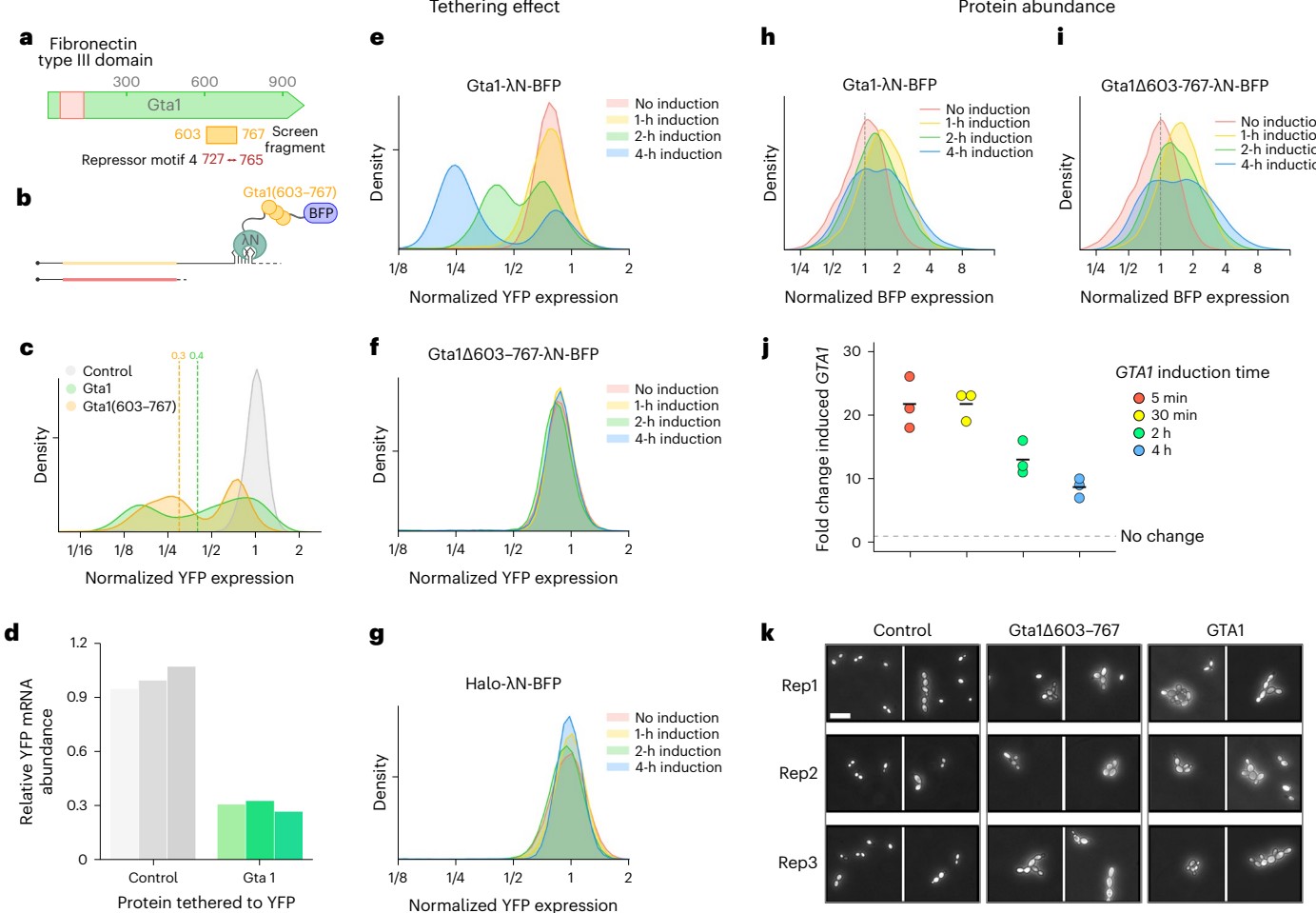

**Fig. 5 | The tethering screen identifies RNA-regulatory roles of poorly characterized proteins. a**, Schematic representation of the Gta1 protein, with the C-terminal Gta1(603–767) fragment highlighted. **b**, Schematic depiction of Gta1(603–767) in the tethering assay. **c**, Flow cytometry measuring activity of Gta1 and Gta1(603–767) in the tethering assay, where dashed lines represent the median YFP expression (*n* = 2, one replicate per sample is shown). **d**, RT–qPCR analysis of YFP mRNA abundance with Gta1 tethered to the 3′ UTR, normalized to a non-regulator control (*n* = 3 independent biological replicates are shown in different shades; *P* = 0.00026, two-sample *t*-test with unequal variance).

**e**–**g**, Time course of reporter changes after induction of Gta1 (**e**), Gta1Δ603–767 (**f**) and Halo control tethering constructs (*n* = 2) (**g**). **h**, Change in BFP fluorescence as a measure of Gta1 expression over time, normalized to the uninduced Gta1 sample (*n* = 2, one replicate is shown per time point). **i**, As in **h**, for Gta1Δ603–767. **j**, RT–qPCR analysis of induced *GTA1* relative to endogenous *GTA1* expression (*n* = 3 biological replicate cultures). **k**, Light microscopy images of yeast overexpressing *GTA1*, *GTA1Δ603–767* or Halo control for 4 h (*n* = 3 biological replicate cultures are shown, with two frames per replicate). Scale bar, 20 μm.

that identified two distinct N-terminal regions required for interactions with translation initiation factors eIF4A and eIF4E[47]. Full activity of Ded1 fragments in the tethered-function assay requires both of these interactions, mediated by Ded1(30–60) and Ded1(60–100), respectively. Our analysis of Ded1 and Ccr4 emphasizes that important regulatory effects are often associated with disordered interaction motifs.

### Regulatory functions of Sro9 and Cdc48

We identified powerful positive regulatory activity in an N-terminal fragment of Sro9 (Fig. 3b,e). This poorly characterized RBP, one of three La-motif-containing proteins in yeast, associates with translating ribosomes[74] and translation initiation factors[35,75]. It appears to bind and stabilize target mRNAs enriched for functions in protein synthesis[35]. Sro9 also contains an activator-associated peptide motif (Fig. 7a and Extended Data Fig. 4b), although our validated N-terminal fragment did not include this motif. We thus tested the full-length protein along with one truncation, Sro9(1–151), that encompassed our validated fragment, and a longer Sro9(1–251) truncation that included the activator-rich motif as well. We also tested the remaining C-terminal

fragment, Sro9(252–434), which includes the La-motif and is implicated in RNA binding[35] (Fig. 7a). Inclusion of the activator-associated motif did not further increase the activity of the Sro9 N terminus, although full-length Sro9 was a substantially stronger activator (Fig. 7b). The C-terminal portion alone was essentially inactive, which is consistent with the separation of RNA-binding regions and effector domains (Fig. 7c). The stronger effect of the full-length protein cannot be explained by differences in protein abundance (Extended Data Fig. 7a), and so the context of full-length Sro9 must potentiate the positive effect of the N-terminal region.

Sro9 is reported to interact with several translation factors, including Pab1, which emerged as a common interaction partner for many activators (Fig. 4d,e)[35]. We thus wanted to test whether the Sro9(1–151) fragment was sufficient for a stable Pab1 interaction. Indeed, we found that Pab1 co-purified with epitope-tagged N-terminal Sro9(1–151) (Fig. 7d and Supplementary Data 7b). The co-purification of Pab1 with full-length Sro9 protein was somewhat stronger than the N terminus, consistent with its stronger activation. Pab1 can enhance expression by stabilizing mRNAs or by promoting their translation. We found

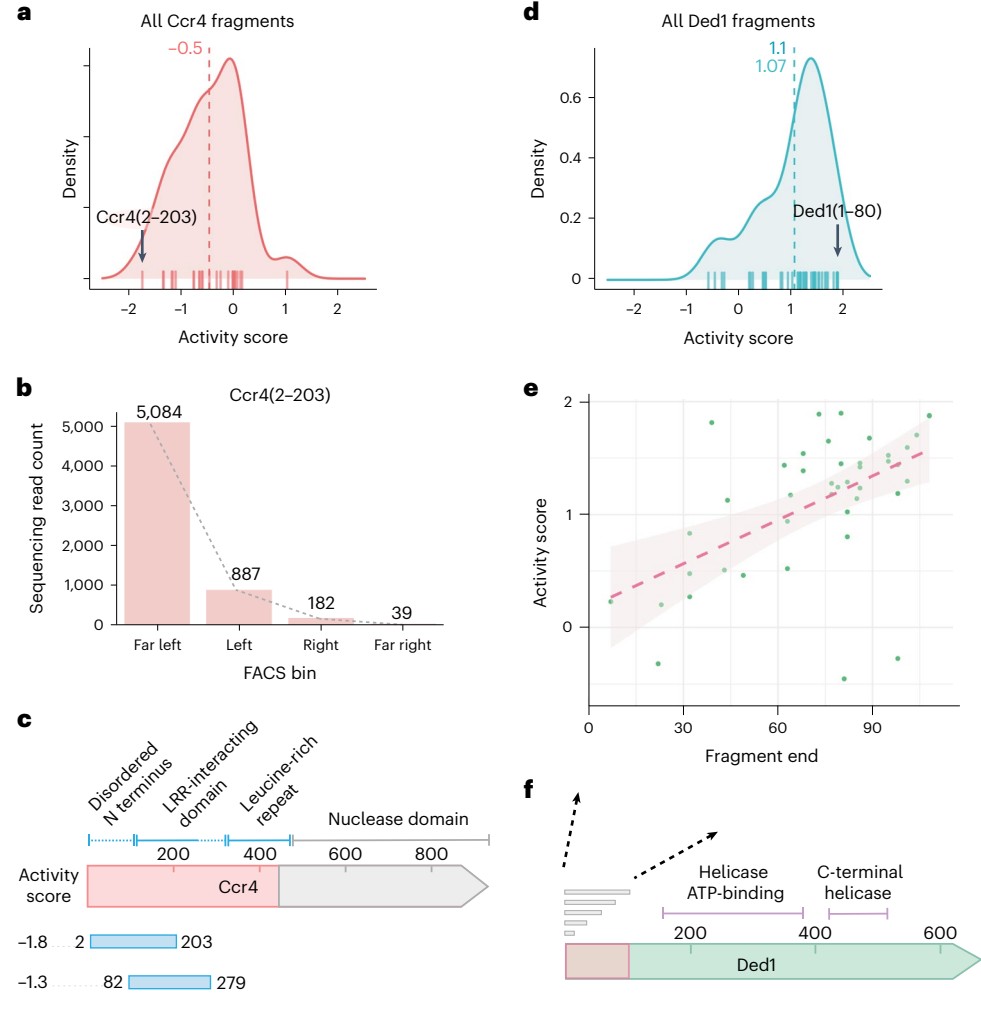

**Fig. 6 | The tethering screen defines functional domain boundaries of well characterized RBPs. a**, KDE for all Ccr4 fragments assayed in the screen. The strongest repressor fragment, Ccr4(2–203), is indicated with a black arrow. **b**, Distribution of read counts per FACS bin for Ccr4(2–203). **c**, Schematic depiction of Ccr4 protein domains, with the strongest repressor fragment from each of the N-terminal domains shown below. **d**, As in **a**, for Ded1. The strongest

activator fragment, Ded1(1–80), is indicated with a black arrow. **e**, Comparison of fragment activity score with the fragment 3′ end; all fragments depicted begin at the endogenous Ded1 initiation site. A linear model fit is shown (dashed line) as well as the 95% confidence interval (shaded). **f**, Schematic depiction of Ded1 protein domains, the fragments spanning the disordered N-terminal 100-amino acid residues are indicated in **e**.

that Sro9 tethering increased YFP mRNA abundance by only ~1.5-fold (Fig. 7e), indicating that increased translation explains the majority of its regulatory effect. These results are consistent with the modest quantitative changes in transcript abundance observed in *sro9Δ* yeast[35].

We also observed positive effects upon tethering proteins with little known role in mRNA regulation. Notably, an N-terminal fragment of the AAA ATPase Cdc48 increased reporter expression, although Cdc48 is linked most prominently with protein degradation, including the ER-associated degradation (ERAD) pathway for quality control of transmembrane and secreted proteins[76]. Cdc48 acts as an unfoldase that extracts proteins from membranes and complexes to make them available for proteasomal degradation[77–80]. Cdc48 was recently reported to interact with RNA[9], although its role in RNA regulation remains mysterious. Cdc48's known functions suggest that it would negatively regulate protein expression.

Nonetheless, tethering the N-terminal Cdc48(1–155) fragment to a reporter transcript robustly activated its expression (Fig. 7f and Supplementary Data 5). Furthermore, this appeared to result from enhanced translation, as reporter mRNA levels increased only modestly

(Fig. 7g). Interestingly, full-length Cdc48 did not show this same activity (Fig. 7f). The N terminus of Cdc48 binds to substrates, cofactors and ubiquitin, while the C-terminal domains form the hexameric AAA ATPase[76] (Fig. 7h). Our results thus implicate cofactor interactions of the isolated N terminus in translational activation. We thus deleted the gene encoding the cofactor Ubx2, which localizes Cdc48 to the sites of ERAD and mitochondrial protein translocation-associated degradation (mitoTAD)[81–83] (Fig. 7i), and saw much weaker Cdc48(1–155) activity (Fig. 7j and Extended Data Fig. 7c). Removal of its UBX-domain, which mediates the interaction between Ubx2 and Cdc48[82], had a smaller impact. Localization of Cdc48(1–155) to endomembranes may recruit tethered transcripts and thereby modulate their expression. Alternately, the isolated N terminus could displace binding of full-length Cdc48 and reduce protein turnover.

## Discussion

We report a broad and unbiased survey of the budding yeast proteome that identifies proteins controlling mRNA translation and decay. We have recovered a wide array of active proteins that includes many known

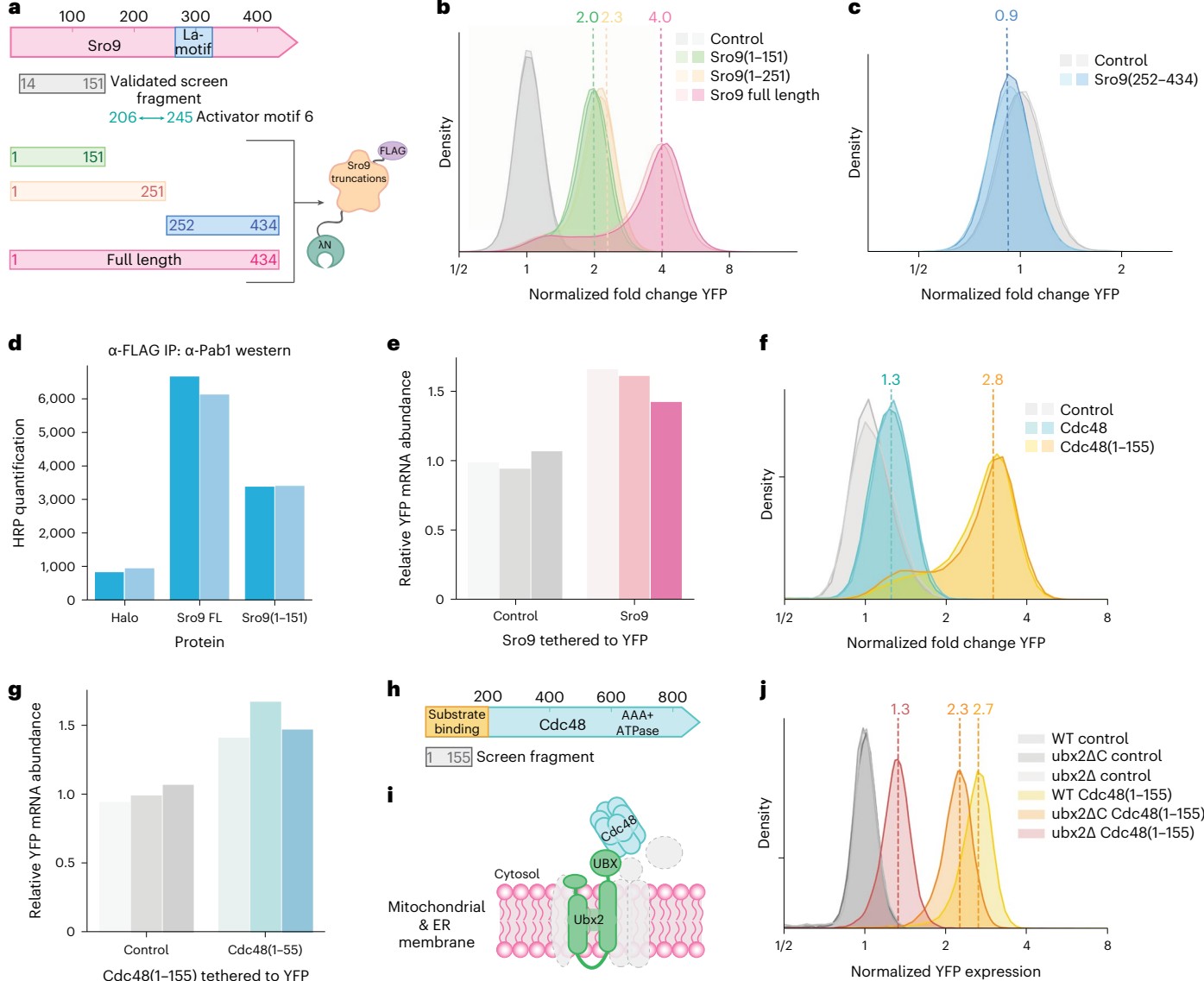

**Fig. 7 | The tethering screen reveals regulatory roles of known RBPs.**
**a**, Schematic representation of Sro9 protein domains and its truncations characterized in the tethering assay. **b**, Flow cytometry measuring activity of Sro9 full length and N-terminal fragments in the tethering assay, where dashed lines represent median YFP expression ($n = 2$ independent biological replicates). **c**, As in **b**, but for C-terminal fragments ($n = 2$ independent biological replicates). **d**, Quantification of horseradish peroxidase (HRP) western blot analysis for Pab1 enrichment in the FLAG-tag immunoprecipitation eluate ($n = 2$ independent biological replicates are shown in different shades). **e**, RT–qPCR analysis of YFP mRNA abundance with Sro9 tethered to the 3′ UTR, normalized to a non-

regulator control ($n = 3$ independent biological replicates are shown in different shades; $P = 0.0013$, two-sample $t$-test with unequal variance). **f**, As in **b**, but for Cdc48 and Cdc48(1–155) ($n = 2$ independent biological replicates). **g**, As in **e**, but for Cdc48(1–155) ($n = 3$ independent biological replicates are shown in different shades; $P = 0.0098$, two-sample $t$-test with unequal variance). **h**, Schematic representation of the Cdc48 protein, with its substrate, cofactor and ubiquitin binding N-terminal domain highlighted. **i**, Schematic depiction of Cdc48 recruitment to the ER and mitochondrial membranes through Ubx2. **j**, As in **b**, for Cdc48(1–155) in the wild-type, *ubx2ΔC* and *ubx2Δ* strains ($n = 2$ independent biological replicates, one replicate per sample is shown).

regulators, strongly enriched for RBPs. We have also delineated the active regions within these proteins, revealing that regulatory activity typically maps outside the RBDs themselves. Post-transcriptional regulators thus seem to display a modular architecture, with RBDs that determine their mRNA specificity and separate regulatory domains that modulate the expression of these target transcripts[5,84]. We have found regulatory activity associated with folded domains, but also with disordered regions, highlighting the importance of functional IDRs in post-transcriptional regulation[84]. Indeed, the repressive fragments of the Ccr4 deadenylase included its disordered N terminus, and disordered fragments from the N terminus of the translation initiation factor Ded1 activated expression. Two broad models have emerged to

explain how such IDRs might show specific molecular functions. General patterns of amino acid composition, such as interspersed acidic and aromatic residues, seem to underlie transcriptional activation by IDRs[85–87]. Other IDRs harbor SLiMs that act through well-defined peptide-protein docking[88]. Because unstructured regions may be easier to capture in our library, or more likely to function in isolation, we could not determine whether activity was more likely to occur in folded domains or disordered regions. Nonetheless, both degenerate and specific peptide motifs emerged from our survey, suggesting that both modes of actions play important roles in mRNA regulation.

Many of the regulators we identified may exert their effect through their interactions with other proteins. This pattern held even in Ccr4

and Ded1, which both contain enzymatic activities that could act directly on a tethered mRNA. Protein-protein interactions can affect expression of a target transcript by recruiting the large, multi-protein complexes involved in translation and mRNA decay or modulating their activity. Indeed, positive regulators were enriched for interaction with the poly-(A) binding protein Pab1, which stabilizes mRNAs and promotes their translation, suggesting that these regulators could converge onto core pathways controlling the fate of mRNAs. Similar patterns have been seen in organisms ranging from trypanosomes to humans, and so this convergence may reflect a general organizational principle of eukaryotic post-transcriptional regulation[16,17].

Although our approach allowed us to take a broad view across the yeast proteome, not restricted to known RBPs, it also necessitated trade-offs. We obtained fragment coverage for roughly half of the yeast proteome. This might in part reflect technical limitations of generating fragments, although we used Tn5 transposase[23], which compares favorably to other methods of random fragmentation. Selection for in-frame fragments could exclude certain proteins based on poor expression or toxicity—although these effects would also interfere with our tethering assay. Additionally, our results reflect regulatory activity on one reporter transcript in a particular growth condition. The regulatory effects of RBPs can vary based on codon optimality[89] and interactions with other regulators binding the same 3′ UTR[90,91]. Understanding how post-transcriptional regulation varies between transcripts and changes in response to cell physiology remains an important challenge.

Despite the limitations, we have identified strong post-transcriptional activity for over a 1,000 protein fragments, laying a foundation for future work. The fluorescence-based tethering assay offers a tool to further explore these regulatory networks, understand the mechanistic basis for post-transcriptional regulation, and decipher the functional consequences of the diverse RNA-binding proteome that has recently come into view.

## Online content

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

## Methods

### Strain construction

The dual fluorescent reporter (YFP::PP7/RFP::boxB) strain NIY289 was constructed as follows. pNTI252 was integrated into BY4741 at URA3 to generate NIY106. pNTI476 was integrated into BY4742 at URA3 to generate NIY287. NIY106 was crossed with NIY287 to create NIY293. The dual fluorescent reporter (YFP::boxB/RFP::PP7) strain NIY293 was constructed as follows. pNTI282 was integrated into BY4741 at URA3 to generate NIY114. pNTI473 was integrated into BY4742 at URA3 to generate NIY286. NIY114 was crossed with NIY286 to create NIY293. Wild-type BY4741 was used for the in-frame library selection. The yKS090 dual reporter strain expressing the ZIF synthetic transcription factor was generated by integrating pNTI727 into the yeast XII-5 integration site[92]. The *UBX2* mutation strains were generated as follows. We amplified the KanMX cassette from pCfB2225 with primers to generate homologous overlapping sequences to the *UBX2* locus in the yeast genome, and then integrated this cassette into one locus of *UBX2* in NIY293. We verified correct chromosomal integration by colony polymerase chain reaction (PCR) which indicated a heterozygous *ubx2Δ*/*UBX2* genotype (yKS092). We then amplified the KILEU2 cassette from pCfB2189 with either homologous sequences to the remaining *UBX2* locus or to the C-terminal UBX domain of *UBX2*, and then integrated these amplicons into yKS092 to create the *ubx2Δ* and *ubx2Δc* strains, respectively (yKS093 and yKS094). Genotypes were confirmed through colony PCR. Plasmids and strains are listed in Supplementary Tables 1 and 2, respectively.

### Culturing conditions

Cultures for the single protein tethering assay were grown to mid-exponential growth phase at an optical density at 600 nm ($OD_{600}$) of 0.6 then collected via gentle centrifugation at 5,000*g* for 1 min for flow cytometry analysis with 30-min incubation in 4% paraformaldehyde (PFA). For in-frame fragment selection, cultures were incubated after transformation at 30 °C with shaking for 96 h in SCD-His medium with twice-daily back-dilution to avoid culture saturation. NIY293 was transformed with the in-frame tethered fragment library using the high-efficiency lithium acetate method[93] and then transferred to a turbidostat[94] for 48 h in SCD-His medium before collecting cells. Cultures for the inducible Gta1 tethering assay were grown to stationary phase overnight, back-diluted to $OD_{600}$ of 0.1 and allowed to reach early exponential growth phase. The tethered proteins were then induced with 5 nM β-estradiol, then collected and fixed in PFA as described above.

### Flow cytometric measurements and fluorescence-activated cell sorting

The expression of YFP and RFP in the tethering assay were measured using flow cytometric readout on a BD LSR Fortessa X20 using BD FACSDiva version 6.2 with excitation by a 488-mm blue laser and 561-mm yellow-green laser, captured on the FITC and PE-TexRed channels, respectively. Fluorescence measurements for 50,000 cells were collected per sample, and gates were drawn to include populations of the ~25% cells with modal forward- and side-scatter. FACS was performed with an Aria Fusion sorter by gating four equal-sized populations based on the ratio of FITC and PE-TexRed emission. Approximately two million cells were sorted into each gate. The sort was performed with two technical replicate libraries from the same library transformation.

### In-frame and fragment tethering library generation

Genomic yeast DNA was tagmented using the Nextera XT DNA library prep kit from Illumina, and then size-selected with Beckman AMPureXP beads. Size selection was confirmed with an Agilent Tapestation 2200 on a High Sensitivity D1000 Screentape (Extended Data Fig. 2a). BY4741 yeast were then co-transformed with the tagmented yeast gDNA and linear pKS132 and cultured as described above. After a long outgrowth in selective media, plasmids were collected with the Zymo yeast miniprep kit. The selected fragments were then amplified by PCR with primers

KS605(GTAATTATCTACTTTTTTACAACAAATcctgcaggGGCTCGGAGATG TGTATAAGAGACAG) and KS630(CTGTCTCTTATACACATCTGACGCcGG AAGCGGAAGCGGAAGCCGCGCCGACGCACAAAC), designed to anneal to the Nextera XT sites introduced during tagmentation, and subcloned into the SbfI-linearized tethering library vector pKS137 by Gibson assembly. This tethering library was propagated in DH10β cells. Barcodes were then introduced by Gibson assembly of N25 randomized oligonucleotide barcodes, amplified with KS633(ACGAGGCGCGTGTAAGTTACAGGC AAGCGATCCGTCCGTAATACGACTCACTATAGCACG) and KS634(GATCC TGTAGCCCTAGACTTGATAGCCATGACTTCAACTCAAGACGCACAGATAT-TATAA) into the BamHI-linearized tethering library. Assembly reactions were transformed into DH10β and selected in liquid cultures at varying dilutions to obtain a transformant pool with approximately three barcodes per fragment. This library was transformed into NIY293 through the lithium acetate method as described in ref. 93. Transformations were used to inoculate a turbidostat and grown in selective SCD-His media for 48 h before performing FACS on live cells. Library plasmid DNA was then collected from sorted cells with the Zymo yeast plasmid prep kit, then barcode RNA was in vitro-transcribed with the HiScribe T7 High Yield RNA kit from New England Biolabs. RNA was collected with the previously described phenol chloroform method[95]. All PCR reactions were performed using Q5 polymerase according to the manufacturer's protocols. Barcodes were amplified through a limited-cycle PCR with Illumina dual-index primers. Barcodes were assigned to yeast fragment DNA with next-generation sequencing using the PacBio single molecule real-time (SMRT) technology (Supplementary Data 2).

### Barcode quantification and sequencing analysis

Sequencing data were processed using cutadapt v1.16 to remove sequencing adapter sequences. HISAT2 v2.1.0 was used to align sequencing reads to the yeast genome to identify fragment DNA. Trimmed barcodes were then counted and tabulated as described in ref. 96. Barcodes that lacked at least 32 counts in one of the sorted gates were filtered out.

### RNA quantification

Total RNA was collected from triplicate cultures of each strain using the phenol chloroform method[95]. Quantification of *YFP* reporter RNA expression in the tethering assay was performed via RT–qPCR analysis by comparing *YFP* Ct values to *RFP* Ct values, and Ct differences for cells expressing an active tethering protein were compared to a tethered Halo protein control Ct differences. The fold change in *GTA1* expression in the induced cultures (Extended Data Fig. 5b) was compared to the endogenous *GTA1* levels in the Halo-expressing control strain.

### Domain and motif enrichment analysis

The search for domain enrichment among the tethered library fragments proceeded as follows. Active fragments were first considered as those with an activity score of less than −1 or more than +1. Active fragments that were 90% or more similar to another fragment were considered overlapping, and the most highly sequenced from a group of overlapping fragments was used in the analysis. A given Pfam protein domain was considered represented if one or more fragments covered at least 75% of the domain. The activity scores of represented protein domains were averaged and the mean value was reported for each domain. False discovery rates were calculated with the Benjamini–Hochberg procedure, and domains with an adjusted *P* value of less than 0.05 are reported in Extended Data Fig. 4c.

To search for short peptide motifs enriched in our active fragments, we again considered active fragments as those with an activity score of less than −1 or greater than +1. We then ran MEME analysis to search for recurring motifs within the sequences of our active fragments[52]. We collapsed sequences that were 50% or more similar into the same fragment to avoid detecting a motif multiple times within the same gene. We then used FIMO[52] to scan the yeast genome for occurrences of the motifs that were enriched in our active fragments.

We manually removed two motifs that came from a single peptide sequence as these did not represent a consensus sequence from multiple distinct proteins, and we removed alignments that fell within highly repetitive genomic sequences.

## Protein expression analysis via western blotting

Total protein was isolated from mid-exponentially growing yeast by rapid capture of protein expression through 5% tricarboxylic acid treatment for 10 min, followed by a wash in acetonitrile. The cell pellets were then dried at room temperature for 30 min before bead-beating in Tris-acetate-EDTA buffer for 5 min at room temperature. Samples were then resuspended in SDS loading buffer from NuPage, boiled for 5 min, and loaded on 4–12% polyacrylamide Bis-Tris gels and separated by electrophoresis in MOPS buffer. Proteins were then transferred to a nitrocellulose membrane and were blocked for 1 h in TBST (Tris-buffered saline, 0.1% Tween 20) with 5% bovine serum albumin. Primary antibodies (DYKDDDDK Tag Antibody, Cell Signaling Technology 2368S; α-Pab1 Antibodies-Online ABIN1580454 (clone 1G1)) were incubated with membranes at a 1:1,000 dilution in TBST for 1 h at room temperature, washed with TBST, then incubated for 30 min at room temperature with anti-rabbit (Cell Signaling Technology, 7074S) and anti-mouse (Cytiva NA931-1ML) HRP-linked antibodies at a 1:10,000 dilution. Membranes were developed with Pierce ECL western blotting substrate and imaged on the chemiluminescence channel on a ProteinSimple instrument.

## Microscopy

Mid-exponential phase cells were collected by gentle centrifugation and then fixed for 30 min in 4% PFA. The cells were washed in 1× PBS buffer and visualized with a Leica DM IL LED microscope at ×40 magnification, acquired with Leica Application Suite v4.8.0, and processed using ImageJ 1.53t. Fields of view for saved images were randomly selected.

## Statistics and reproducibility

No statistical method was used to predetermine sample sizes, but our sample sizes are similar to those reported in previous publications (ref. 97). No data were excluded from the analyses, and all replicate experiments were successful. The experiments were not randomized. The investigators were not blinded to allocation during experiments and outcome assessment. Data distributions were assumed to be normal, but this was not formally tested.

## Reporting summary

Further information on research design is available in the Nature Portfolio Reporting Summary linked to this Article.

## Data availability

High-throughput sequencing data has been deposited with the NCBI Short Read Archive. Long-read sequencing linking tethering constructs and barcodes is available at SRR10355648, and short-read sequencing quantifying the barcodes is available at SRR10353306 through SRR10353315, as described below.

| Accession | Title |
|---|---|
| SRR10355648 | Tethering construct library |
| SRR10353306 | Tethering construct barcode, unsorted, replicate 2 |
| SRR10353307 | Tethering construct barcode, far-right sort, replicate 2 |
| SRR10353308 | Tethering construct barcode, near-right sort, replicate 2 |
| SRR10353309 | Tethering construct barcode, near-left sort, replicate 2 |
| SRR10353310 | Tethering construct barcode, far-left sort, replicate 2 |
| SRR10353311 | Tethering construct barcode, unsorted, replicate 1 |
| SRR10353312 | Tethering construct barcode, far-right sort, replicate 1 |
| SRR10353313 | Tethering construct barcode, near-right sort, replicate 1 |
| SRR10353314 | Tethering construct barcode, near-left sort, replicate 1 |
| SRR10353315 | Tethering construct barcode, far-left sort, replicate 1 |

Publicly available datasets used here include the *S. cerevisiae* proteome and Pfam domain annotations from InterPro proteome UP000002311, and *S. cerevisiae* genome annotations from http://sgd-archive.yeast-genome.org/sequence/S288C_reference/genome_releases/S288C_reference_genome_R64-2-1_20150113.tgz. BioGRID data are from https://downloads.thebiogrid.org/Download/BioGRID/Release-Archive/BIOGRID-4.3.195/BIOGRID-MV-Physical-4.3.195.tab3.zip. Source data are provided with this paper.

## Code availability

Custom software and scripts are available from Zenodo at https://doi.org/10.5281/zenodo.4963329.

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

## Acknowledgements

We thank S. Iwasaki and G. Brar for insightful comments, R. Muller (UC Berkeley) for library plasmids, S. Fernandez for assistance with computational data visualization, and J. Lobel along with other members of the Ingolia laboratory for thoughtful scientific discussions. This work was supported by NIH grants DP2 CA195768 (N.T.I.) and R01 GM130996 (N.T.I.) and by the Rose Hills Innovator Program, the Vincent J. Coates Genomics Sequencing Laboratory at UC Berkeley, supported by NIH S10 OD018174 Instrumentation Grant, and the Flow Cytometry Facility at UC Berkeley, the UC Berkeley DNA Sequencing Facility and the UC Davis Proteomics Core Facility.

## Author contributions

K.R., A.M. and N.I. conceived and designed the experiments. K.R. and A.M. carried out experiments with assistance from Z.M. K.R., N.I. and D.N. analyzed the data. N.I. supervised the project. K.R. and N.I. drafted the manuscript, with revisions from A.M.

## Competing interests

N.I. declares financial interests in Velia Therapeutics and Tevard Biosciences. The other authors declare no competing interests.

## Additional information

**Extended data** is available for this paper at https://doi.org/10.1038/s41594-023-00999-5.

**Correspondence and requests for materials** should be addressed to
Nicholas T. Ingolia.

**Peer review information** *Nature Structural & Molecular Biology*
thanks Gian Gaetano Tartaglia and the other, anonymous,
reviewer(s) for their contribution to the peer review of this work.

Primary Handling Editor: Carolina Perdigoto, in collaboration with
the *Nature Structural & Molecular Biology* team. Peer reviewer reports
are available.

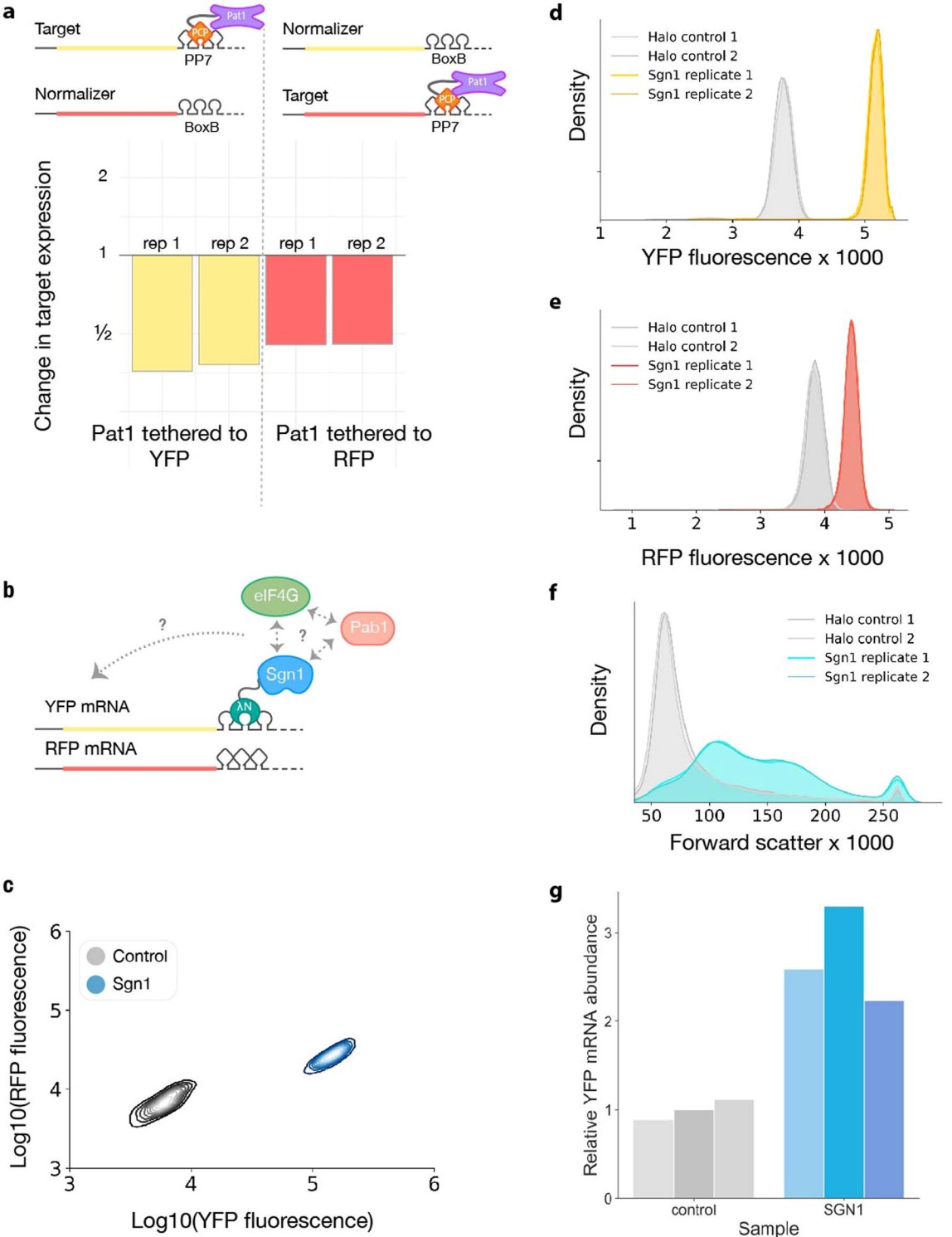

**Extended Data Fig. 1 | The dual reporter tethering assay reports reproducible and quantitative regulatory effects. a**, Pat1 activity tethered to 3′ UTR of both fluorescent reporters with PP7 is reproducible between replicates and fluorophores. **b**, Schematic representation of Sgn1 recruiting Pab1 and eIF4G in the tethering assay. **c**, Comparison of RFP and YFP fluorescence with Sgn1 or a non-regulator control tethered to YFP (n = 3, one representative replicate depicted). **d**, YFP fluorescence with Sgn1 and the non-regulator Halo protein tethered to the 3′ UTR. **e**, RFP fluorescence with Sgn1 and the non-regulator Halo protein tethered to the 3′ UTR of YFP. **f**, Forward scatter fluorescence of Sgn1 and control protein expressing cells indicates larger cell size in the Sgn1 samples. **g**, RT-qPCR analysis of YFP mRNA levels with Sgn1 or the control tethered to the 3′ UTR.

**a** Fragment input library size distribution:

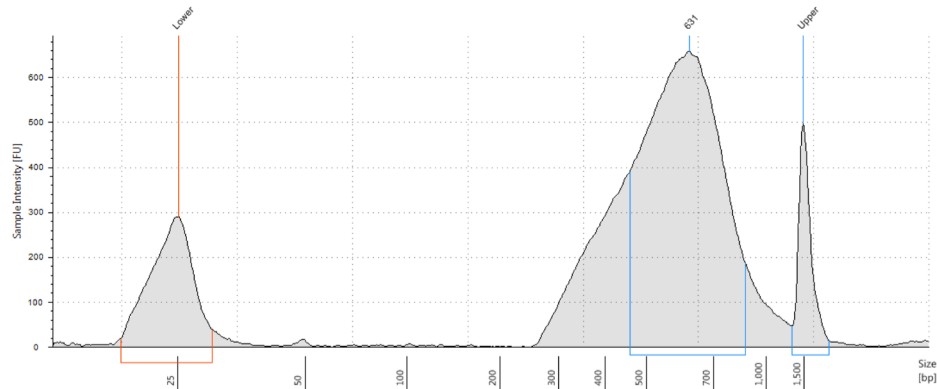

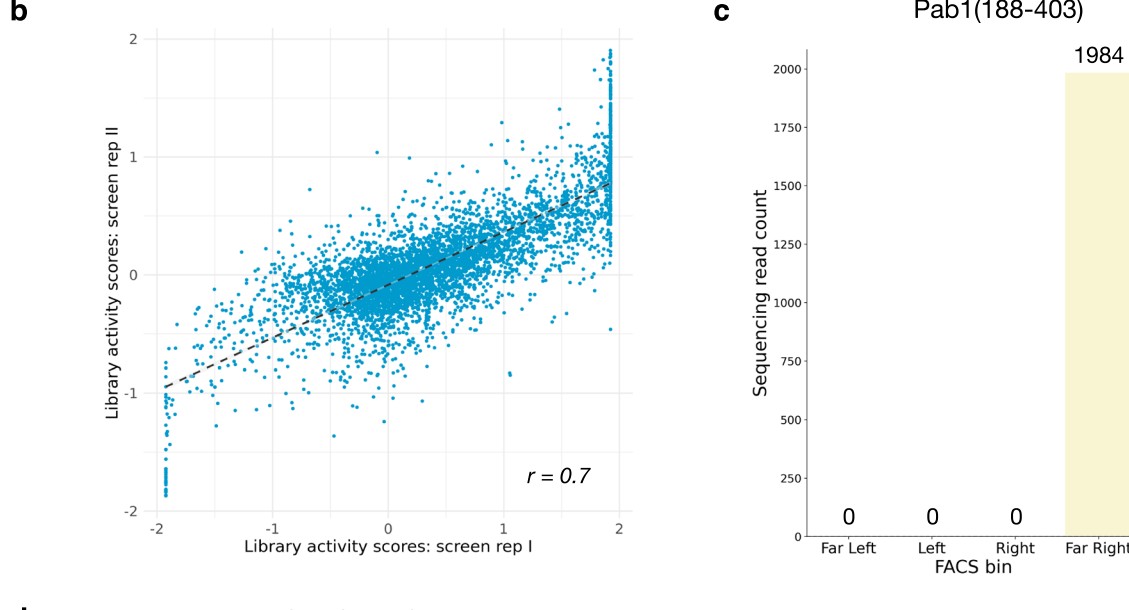

**b**

**c** Pab1(188–403)

**d** Cth1(38–91)

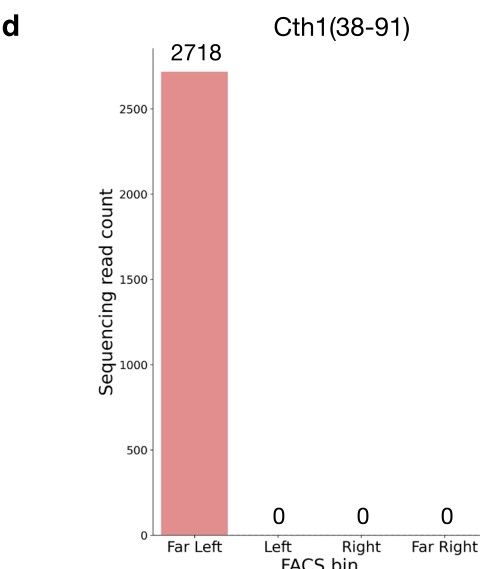

**Extended Data Fig. 2 | Generating an unbiased, proteome-wide survey of tethered in-frame protein fragments. a**, Bioanalyzer analysis of tagmented genomic DNA library size distribution. **b**, As in Fig. 2e, for Pab1(188–403). **c**, as in Fig. 2e, for Cth1(38–91). **d**, Comparison of library-wide activity scores between screen replicates I and II (r = 0.7).

**a**

| barcode | ORF | gene | fragment start | far left counts | left counts | right counts | far right counts | unsorted counts | activity score |
|---|---|---|---|---|---|---|---|---|---|
| GGATACTAGCGCTCTGGTCATCTTA | YDR206W | EBS1 | chr04:864649- | 1178 | 0 | 3 | 2 | 10 | -1.90 |
| TGTGTTGTGGGTAGTTATTGAATAG | YBR072W | HSP26 | chr02:382528- | 2232 | 50 | 93 | 9 | 29 | -1.71 |
| CGCCTTTATTGGGTTTATCTGTGGG | YBR212W | NGR1 | chr02:649731- | 1508 | 186 | 8 | 5 | 21 | -1.62 |
| CGGTGCGTATTTCGGCTCGGTGTGC | YJR091C | JSN1 | chr10:596252+ | 427 | 2 | 98 | 5 | 6 | -1.28 |
| GGTGCACCATCCCGGACTATCGTG | YHR161C | YAP1801 | chr08:420484+ | 2011 | 710 | 13 | 175 | 43 | -1.10 |
| GGGGTAGGTACGAATAGACCCGTGG | YBR172C | SMY2 | chr02:580677+ | 10527 | 2584 | 1697 | 420 | 294 | -1.09 |
| GCGGGATCCACTGGGCGGCGAGATA | YOR227W | HER1 | chr15:766230- | 64 | 272 | 144 | 430 | 17 | 0.63 |
| TTTCTGATTGACGCTAAGCTTACTG | YNL197C | WHI3 | chr14:267662+ | 5 | 493 | 510 | 995 | 30 | 0.82 |
| GATAACACTTGCAGACCAATCTTTT | YDR429C | TIF35 | chr04:1324585+ | 46 | 1243 | 1136 | 2637 | 66 | 0.84 |
| ACCGGGTTTTAAATTTTCTAAATCA | YCL037C | SRO9 | chr03:58019+ | 239 | 5 | 36 | 2101 | 36 | 1.49 |
| AAGTTCGGTCTGCACTAACCGTAAT | YOR204W | DED1 | chr15:723124- | 5 | 0 | 10 | 1525 | 22 | 1.90 |
| TCTTCACTTACGGTGGAGTGGAATT | YHL034C | SBP1 | chr08:33544+ | 0 | 0 | 0 | 1553 | 690 | 1.92 |

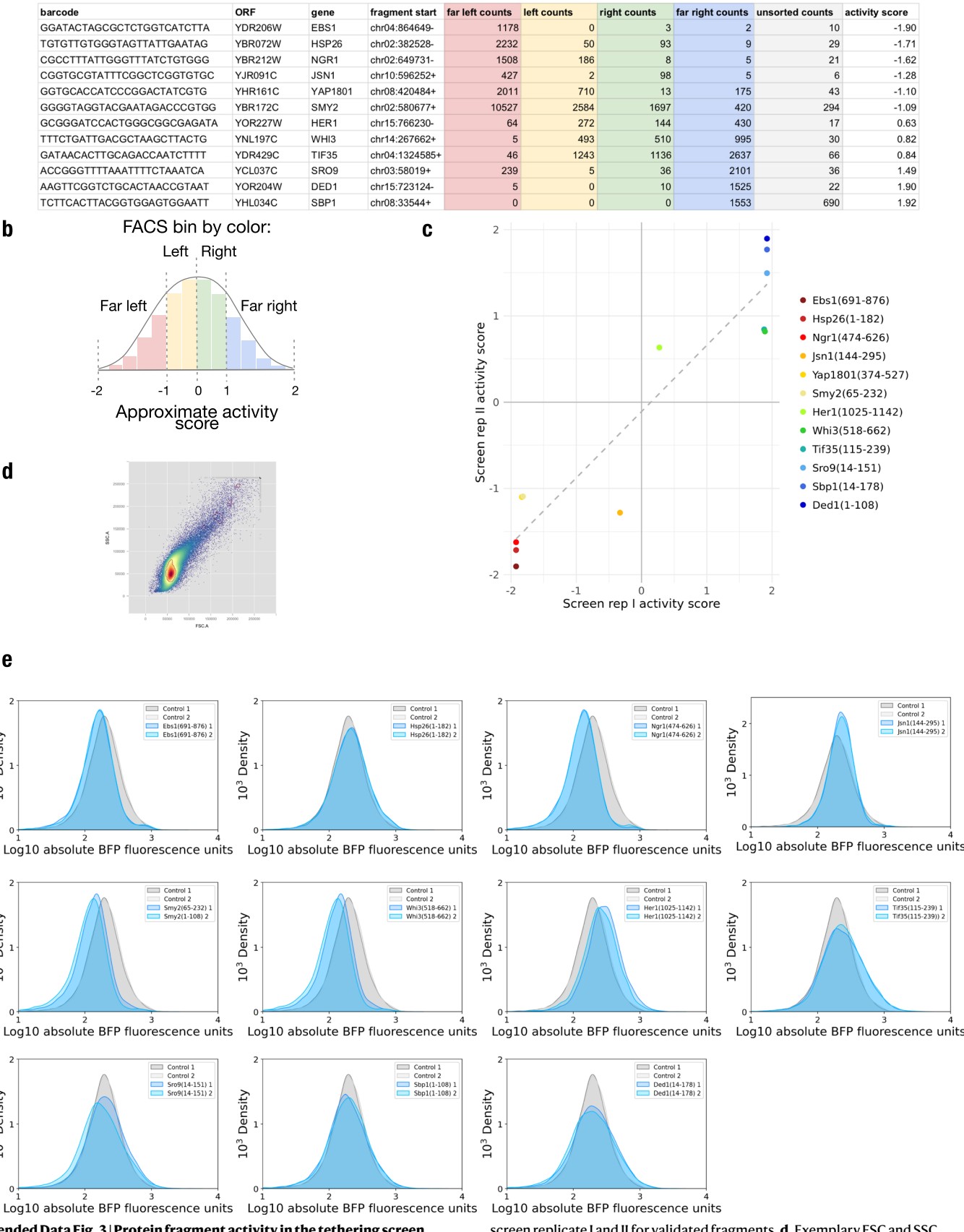

**b** FACS bin by color:

**c**

**d**

**e**

**Extended Data Fig. 3 | Protein fragment activity in the tethering screen represents real, verifiable regulatory function. a**, Sequencing read counts and activity scores for validated screen fragments. **b**, Schematic key depicting FACS bins based on position and color, **c**, Comparison of activity scores between screen replicate I and II for validated fragments, **d**, Exemplary FSC and SSC data showing gate in red, **e**, Absolute BFP fluorescence measurements by flow cytometry showing expression of validated fragment and BFP fusion proteins.

**a**

| | Repressors: Enriched motifs | Motif E-value | Most significant gene | Gene q-value | Count |
|---|---|---|---|---|---|
| 1 |  | $7.3 \times 10^{-88}$ | YKR102W, FLO10 | $3.39 \times 10^{-8}$ | 106 |
| 2 |  | $1.8 \times 10^{-14}$ | YFR008W, FAR7 | $4.26 \times 10^{-10}$ | 524 |
| 3 |  | $2.7 \times 10^{-14}$ | YML010W, SPT5 | $5.48 \times 10^{-05}$ | 1 |
| 4 |  | $1.1 \times 10^{-6}$ | YEL043W, GTA1 | $2.06 \times 10^{-43}$ | 3 |
| 5 |  | $5.9 \times 10^{-6}$ | YFL016C, MDJ1 | $7.41 \times 10^{-05}$ | 9 |
| 6 |  | $1.4 \times 10^{-4}$ | YKR102W, FLO10 | $9.22 \times 10^{-13}$ | 483 |

**b**

| | Activators: Enriched motifs | Motif E-value | Most significant gene | Gene q-value | Count |
|---|---|---|---|---|---|
| 1 |  | $1.2 \times 10^{-513}$ | YKR103W, PRY2 | $1.06 \times 10^{-3}$ | 157 |
| 2 |  | $8.4 \times 10^{-69}$ | YOL155C, HPF1 | $2.04 \times 10^{-25}$ | 563 |
| 3 |  | $1.7 \times 10^{-54}$ | YJR151C, DAN4 | $1.59 \times 10^{-20}$ | 38 |
| 4 |  | $2.5 \times 10^{-41}$ | YDR150W, NUM1 | $1.61 \times 10^{-25}$ | 9 |
| 5 |  | $6.6 \times 10^{-37}$ | YDL014W, NOP1 | $2.12 \times 10^{-12}$ | 63 |
| 6 |  | $2.3 \times 10^{-35}$ | YCL037C, SRO9 | $5.31 \times 10^{-36}$ | 618 |

**Extended Data Fig. 4 | Motifs enriched amongst most active screen fragments. a**, Peptide motifs significantly enriched amongst repressor screen fragments. Counts represent significant occurrences of that motif in the yeast genome. **b**, As in **a**, for the activator screen fragments.

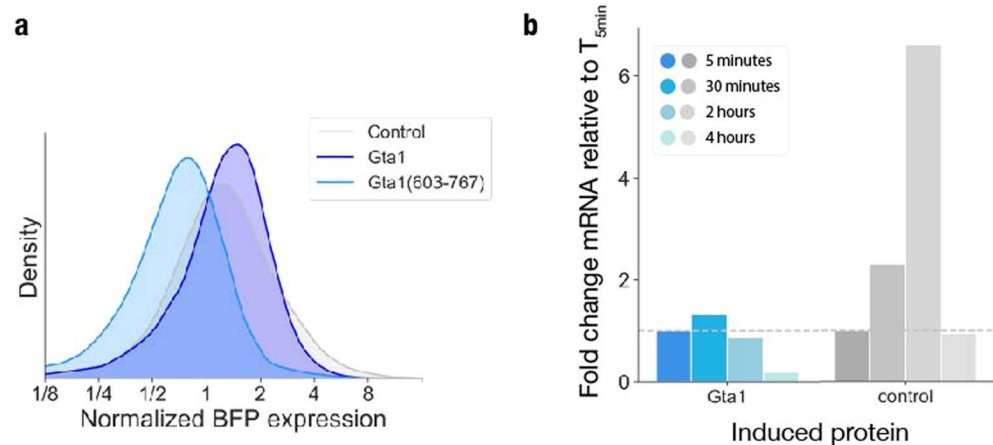

**Extended Data Fig. 5 | The tethering screen identifies RNA-regulatory roles of poorly-characterized proteins. a**, Histogram of BFP fluorescence as a measure for control, Gta1 and Gta1(603–767) expression and stability in the tethering assay, normalized to control BFP levels. **b**, RT-qPCR analysis of induced GTA1 and control mRNA over time, normalized to expression levels at 5 minutes induction.

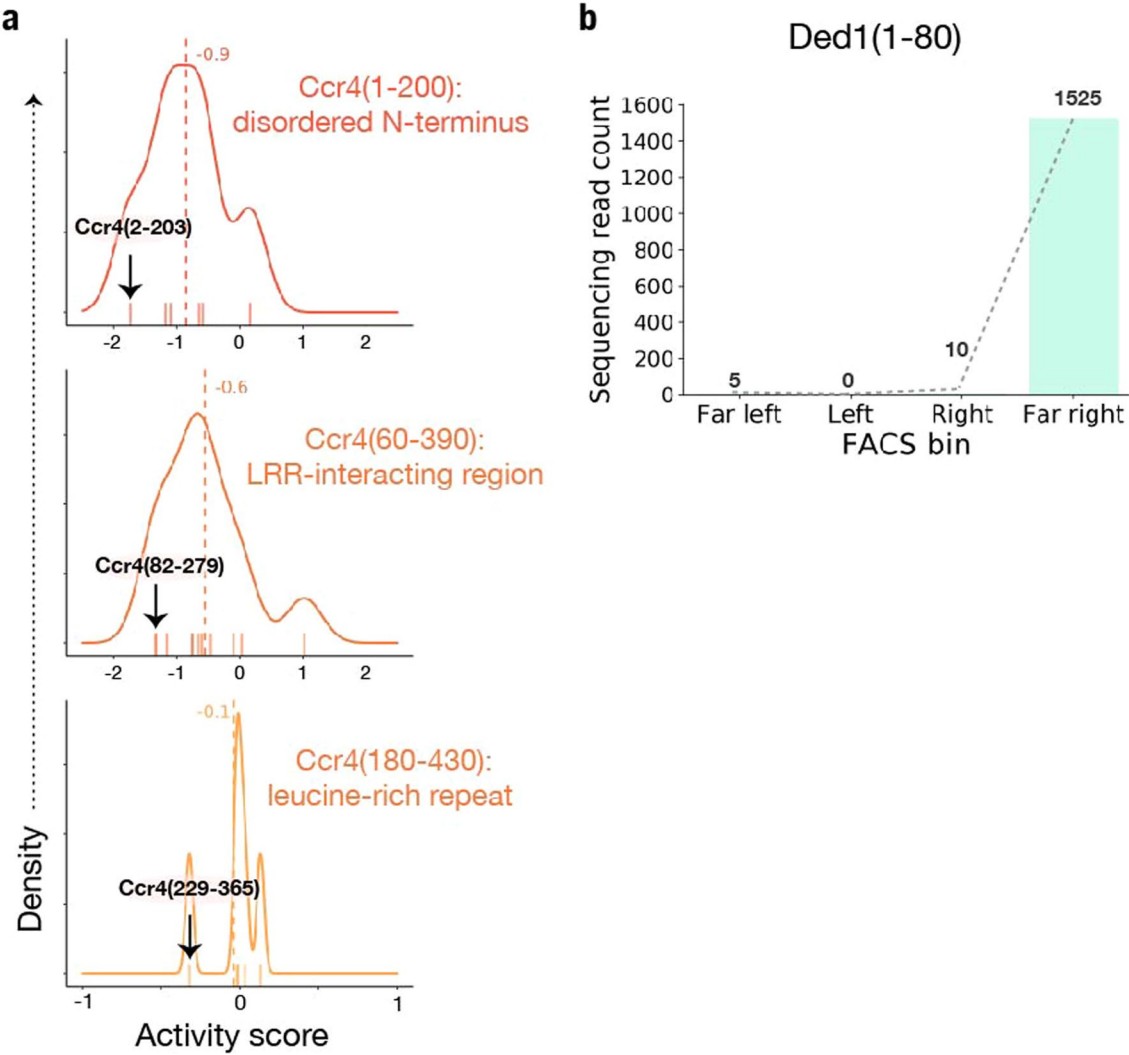

**Extended Data Fig. 6 | The tethering screen defines functional domain boundaries of well characterized RBPs. a**, As in Fig. 6a, for fragments derived from Ccr4(1–200) disordered N-terminus domain (top), Ccr4(60–390) Leucine-rich repeat interacting domain (middle), and Ccr4(180–340) leucine-rich repeat domain (bottom). **b**, As in Fig. 6b, for Ded1(1–80).

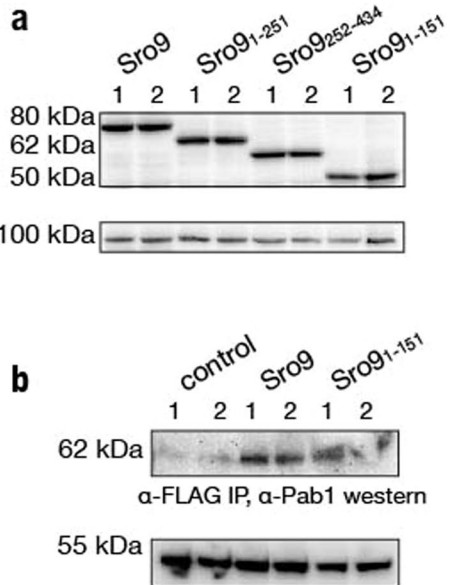

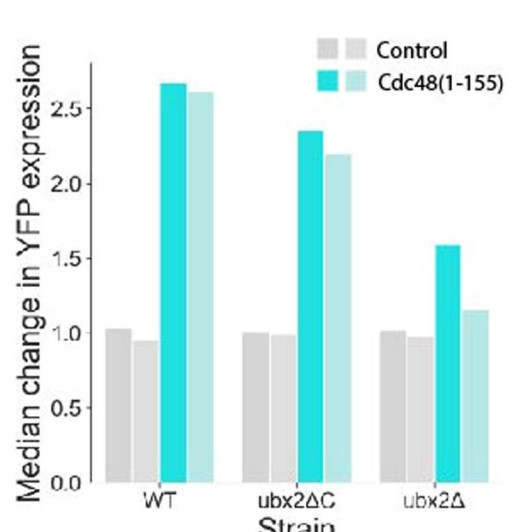

**Extended Data Fig. 7 | The tethering screen reveals regulatory roles of known RNA-binding proteins. a**, Western blot analysis of Sro9 full length and truncation protein expression in the tethering assay. Two independent biological replicates are shown. **b**, Western blot analysis of Pab1 enrichment in FLAG-tag protein purification eluate. Two independent biological replicates are shown. **c**, Quantification of median values in Fig. 7j for two biological replicates.

# Reporting Summary

## Statistics

For all statistical analyses, confirm that the following items are present in the figure legend, table legend, main text, or Methods section.

| n/a | Confirmed | |
|---|---|---|
| ☐ | ☒ | The exact sample size (*n*) for each experimental group/condition, given as a discrete number and unit of measurement |
| ☐ | ☒ | A statement on whether measurements were taken from distinct samples or whether the same sample was measured repeatedly |
| ☐ | ☒ | The statistical test(s) used AND whether they are one- or two-sided<br>*Only common tests should be described solely by name; describe more complex techniques in the Methods section.* |
| ☒ | ☐ | A description of all covariates tested |
| ☐ | ☒ | A description of any assumptions or corrections, such as tests of normality and adjustment for multiple comparisons |
| ☐ | ☒ | A full description of the statistical parameters including central tendency (e.g. means) or other basic estimates (e.g. regression coefficient) AND variation (e.g. standard deviation) or associated estimates of uncertainty (e.g. confidence intervals) |
| ☐ | ☒ | For null hypothesis testing, the test statistic (e.g. *F*, *t*, *r*) with confidence intervals, effect sizes, degrees of freedom and *P* value noted<br>*Give P values as exact values whenever suitable.* |
| ☒ | ☐ | For Bayesian analysis, information on the choice of priors and Markov chain Monte Carlo settings |
| ☒ | ☐ | For hierarchical and complex designs, identification of the appropriate level for tests and full reporting of outcomes |
| ☐ | ☒ | Estimates of effect sizes (e.g. Cohen's *d*, Pearson's *r*), indicating how they were calculated |

*Our web collection on statistics for biologists contains articles on many of the points above.*

## Software and code

Policy information about availability of computer code

| Data collection | BD FACSDiva Software Version 6.2; Leica Application Suite v4.8.0 |
|---|---|
| Data analysis | ImageJ 1.53t<br>cutadapt version 1.16<br>HISAT2 version 2.1.0<br>bedtools v2.25.0<br>Custom and open source software used to analyze data in this study are provided in Zenodo at https://zenodo.org/record/6496503 |

For manuscripts utilizing custom algorithms or software that are central to the research but not yet described in published literature, software must be made available to editors and reviewers. We strongly encourage code deposition in a community repository (e.g. GitHub). See the Nature Portfolio guidelines for submitting code & software for further information.

## Data

Policy information about availability of data

All manuscripts must include a data availability statement. This statement should provide the following information, where applicable:
- Accession codes, unique identifiers, or web links for publicly available datasets
- A description of any restrictions on data availability
- For clinical datasets or third party data, please ensure that the statement adheres to our policy

High-throughput sequencing data has been deposited with the NCBI Short Read Archive. Long-read sequencing linking tethering constructs and barcodes is available at SRR10355648, and short-read sequencing quantifying the barcodes is available at SRR10353306 through SRR10353315, as described below.

# Field-specific reporting

Please select the one below that is the best fit for your research. If you are not sure, read the appropriate sections before making your selection.

☒ Life sciences  ☐ Behavioural & social sciences  ☐ Ecological, evolutionary & environmental sciences

For a reference copy of the document with all sections, see nature.com/documents/nr-reporting-summary-flat.pdf

# Life sciences study design

All studies must disclose on these points even when the disclosure is negative.

| | |
| --- | --- |
| Sample size | No sample size calculations were performed, and sample sizes were based on previous studies (Muller et al., 2020; Reynaud et al., 2021). Experiments were carried out with biological duplicates or triplicates as described in the text, following approaches commonly accepted for high-throughput genome-wide experiments. |
| Data exclusions | No data were excluded. |
| Replication | Experiments were conducted using biological duplicate or triplicate samples as indicated and all replication was successful. |
| Randomization | Randomization was not relevant to experimental designs in this study: no prospective assignment was performed, control and experimental samples were processed in parallel, and data acquisition and analysis were automated and applied uniformly across all samples. |
| Blinding | Blinding was not relevant to this study: observer bias is not relevant because data acquisition and analysis are automated and applied uniformly for all samples. |

# Reporting for specific materials, systems and methods

We require information from authors about some types of materials, experimental systems and methods used in many studies. Here, indicate whether each material, system or method listed is relevant to your study. If you are not sure if a list item applies to your research, read the appropriate section before selecting a response.

## Materials & experimental systems

| n/a | Involved in the study |
| --- | --- |
| ☐ | ☒ Antibodies |
| ☒ | ☐ Eukaryotic cell lines |
| ☒ | ☐ Palaeontology and archaeology |
| ☒ | ☐ Animals and other organisms |
| ☒ | ☐ Human research participants |
| ☒ | ☐ Clinical data |
| ☒ | ☐ Dual use research of concern |

## Methods

| n/a | Involved in the study |
| --- | --- |
| ☒ | ☐ ChIP-seq |
| ☐ | ☒ Flow cytometry |
| ☒ | ☐ MRI-based neuroimaging |

## Antibodies

| | |
| --- | --- |
| Antibodies used | Cell Signaling Technology 2368S DYKDDDDK Tag Antibody (Binds to same epitope as Sigma's Anti-FLAG(R) M2 Antibody) (lot 12) α-Pab1 Antibodies-Online ABIN1580454 (clone 1G1) |

anti-rabbit IgG, HRP-linked (Cell Signaling Technology 7074S, lot 31)
anti-mouse IgG, HRP-linked (Cytiva NA931-1ML)

| Validation | FLAG/DYKDDDDK epitope tag is widely used to recognize epitope-tagged transgenes in S. cerevisiae (Liu et al., Nat Commun 12: 57 (2021); Sun et al., Cell Reports 36(12): 109717 (2021)). We validate the specificity of this tag by detecting four distinct transgenes at their respective, predicted sizes (Extended Data Fig. 7a). Pab1 antibody is a monoclonal antibody (clone 1G1) previously shown to recognize S. cerevisiae Pab1 by mRNA binding in RNA Interactome Capture that can be competed with free poly-(A) RNA (Matia-Gonzalez et al., STAR Protoc 2021). |
| --- | --- |

# Flow Cytometry

## Plots

Confirm that:

☒ The axis labels state the marker and fluorochrome used (e.g. CD4-FITC).

☒ The axis scales are clearly visible. Include numbers along axes only for bottom left plot of group (a 'group' is an analysis of identical markers).

☒ All plots are contour plots with outliers or pseudocolor plots.

☒ A numerical value for number of cells or percentage (with statistics) is provided.

## Methodology

| Sample preparation | Budding yeast were grown in batch culture or turbidostat in mid-exponential growth. Flow cytometry used cells fixed for 30 minutes in 4% PFA Cell sorting used live yeast in 1x PBS All flow cytometry experiments measured genetically encoded fluorescent proteins expressed transgenically |
| --- | --- |
| Instrument | BD LSR Fortessa X20 and Aria Fusion sorter |
| Software | BD FACSDiva Software Version 6.2 |
| Cell population abundance | Fluorescence measurements for 50,000 cells were collected per sample, and gates were drawn to include populations of the ~25% cells with modal forward- and side-scatter |
| Gating strategy | Fluorescence measurements for 50,000 cells were collected per sample, and gates were drawn to include populations of the ~25% cells with modal forward- and side-scatter Fluorescence activated cell sorting was performed with an Aria Fusion sorter by gating four equal-sized populations based on the ratio of FITC and PE-TexRed emission. Approximately two million cells were sorted into each gate. The sort was performed with two technical replicate libraries from the same library transformation. Gating strategy is exemplified in Extended Data Fig. 3b and 3d |

☒ Tick this box to confirm that a figure exemplifying the gating strategy is provided in the Supplementary Information.

