## [Peer Review File · Nature Structural & Molecular Biology]

Peer Review Information

Manuscript Title: Surveying the global landscape of post-transcriptional regulators

Corresponding author name(s): Nicholas Ingolia

Reviewer Comments & Decisions:

Decision Letter, initial version:
--

Message: 8th Oct 2021

Dear Dr. Ingolia,

Thank you again for submitting your manuscript "Surveying the global landscape of post-transcriptional regulators". I apologize for the delay in responding, which resulted from the difficulty in obtaining suitable referee reports. Nevertheless, we now have comments (below) from the 3 reviewers who evaluated your paper. In light of those reports, we remain interested in your study and would like to see your response to the comments of the referees, in the form of a revised manuscript.

You will see that while all three referees appreciate the potential novel insights provided by your approach, they raise some important concerns, particularly regarding using fragment proteins and library coverage, which we expect to see addressed in the revised study. Please also be sure to address to all other concerns of the referees in full in a point-by-point response and highlight all changes in the revised manuscript text file. If you have comments that are intended for editors only, please include those in a separate cover letter.

We expect to see your revised manuscript within 6 weeks. If you cannot send it within this time, please contact us to discuss an extension; we would still consider your revision, provided that no similar work has been accepted for publication at NSMB or published elsewhere.

As you already know, we put great emphasis on ensuring that the methods and statistics reported in our papers are correct and accurate. As such, if there are any changes that should be reported, please submit an updated version of the Reporting Summary along

with your revision.

Reporting Summary:

Please note that all key data shown in the main figures as cropped gels or blots should be presented in uncropped form, with molecular weight markers. These data can be aggregated into a single supplementary figure item. While these data can be displayed in a relatively informal style, they must refer back to the relevant figures. These data should be submitted with the final revision, as source data, prior to acceptance, but you may want to start putting it together at this point.

Data availability: this journal strongly supports public availability of data. All data used in accepted papers should be available via a public data repository, or alternatively, as Supplementary Information. If data can only be shared on request, please explain why in your Data Availability Statement, and also in the correspondence with your editor. Please note that for some data types, deposition in a public repository is mandatory - more

information on our data deposition policies and available repositories can be found below:
<https://www.nature.com/nature-research/editorial-policies/reporting-standards#availability-of-data>

Nature Structural & Molecular Biology is committed to improving transparency in authorship. As part of our efforts in this direction, we are now requesting that all authors identified as 'corresponding author' on published papers create and link their Open Researcher and Contributor Identifier (ORCID) with their account on the Manuscript Tracking System (MTS), prior to acceptance. This applies to primary research papers only. ORCID helps the scientific community achieve unambiguous attribution of all scholarly contributions. You can create and link your ORCID from the home page of the MTS by clicking on 'Modify my Springer Nature account'. For more information please visit please visit www.springernature.com/orcid.

[Redacted]

Sincerely,

Carolina

Carolina Perdigoto, PhD
Chief Editor
Nature Structural & Molecular Biology
orcid.org/0000-0002-5783-7106

Referee expertise:

Referee #1: mRNA decay and translation

Referee #2: RNAPs and proteomics

Referee #3: RNAPs and method development

Reviewers' Comments:

Reviewer #1:

Remarks to the Author:

In this manuscript, Reynaud et al use an mRNA tethering assay to survey the ability for sections of yeast proteins to alter post-transcriptional regulation. A similar scheme has been used to assay the impact of ~700 RNA binding proteins (RBPs) on mRNA decay and translation in human cells, and a step forward in the current manuscript is that the authors (1) do not restrict themselves to known RBPs by generating their fragment library in an unbiased fashion, and (2) investigate the impact of different portions of individual proteins. The dataset made by the authors is a hypothesis-generating dataset that will be of interest. The authors also used a clever approach to generate their fragment library that, again, could be useful for the community. However, in my opinion, there are substantial weaknesses in the current study that lead me to have significant reservations:

(1) As a general statement, without the corresponding library of full-length proteins, it is very hard to interpret the impact of different fragments and their biological relevance. For example, in figure 6, it's hard to interpret the effect of different fragments for Ccr4p and Ded1p without showing all the fragments and whole protein.

(2) One challenge in using protein fragments generated by a shotgun method is that, as the authors know, there is a nontrivial potential for the fragments to have altered folding, expression level, localization, etc.—these effects introduce numerous false negatives and positives into the dataset, which make some of the general interpretations about enrichment fraught with alternative explanations. For example, perhaps unstructured regions are easier to express (or function) than structured domains when the fragments are generated this way, and that causes the enrichments seen by the authors. Similarly, for fragments from nuclear proteins, what are their absolute activity scores as opposed to ones from cytoplasmic proteins? The lack of control experiments and exploring alternative hypotheses, even for a subset of fragments, make interpretation of the data challenging.

(3) Another major caveat comes in terms of the library size. The authors state "We adapt the tethering assay to survey regulatory activity across the entire yeast proteome in an unbiased manner," but it is unclear how many genes are represented in the library, the fraction of each gene covered, the coverage for represented amino acids, etc. Similarly, given that the authors use a genomic shotgun approach, it would be useful to discuss false positives in terms of fragments generated from noncoding regions of the genome? How many of these are represented in the final library? Are any effects observed? A more thorough discussion and analysis of the final library would help substantially.

(4) How many replicates of the screen were performed? How reproducible were the scores? As far as I could see, I could not see this described.

(5) How many fragments did the authors call as having a high absolute activity score? How many genes were represented? How did this change with different thresholds?

(6) In general, some of the follow up experiments (e.g., the GTA1 section) feel underdeveloped and do not add much to the manuscript.

Minor comments:

(1) "Our screen identified strong activity in fragments lacking an identifiable, folded domain." Example data here or a pointer to the results would be useful.

Reviewer #2:

Remarks to the Author:

In this manuscript, Reynaud et al designed and employed a screening approach to identify proteins and their domains that modulate post-transcriptional regulation of mRNA in yeast. They achieve this by tethering randomly-generated fragments of genomic DNA to a reporter system, consisting of a YFP reporter whose activity was read by FACS sorting. In particular, the increase or decrease of YFP relative to RFP in the control was used to indicate translational activation or repression, respectively. Identity of regulatory fragments was performed by deep sequencing, and aggregate data were investigated to reveal global patterns in proteins and domains that were recovered. In addition, selected candidates were examined, identifying examples where unstructured domains outside RNA-interaction domains contributed to regulation of translation. The authors imply a network of proteins for post-transcriptional gene regulation.

The paper introduces an interesting and potentially powerful approach to identify genes and gene-fragments involved in post-transcriptional gene regulation, by tethering baits of interest (either in a library or in a targeted fashion) to a reporter system. Here, a tagged and size-selected genomic library was generated to span the coding yeast genome. The study convinces in demonstrating that this system can be used to identify known and novel protein domains with regulatory activity in post-transcriptional gene expression, however it is weaker in other aspects, especially with regard to indicating overall relevance for such activity in full-length proteins, in explaining observed bimodal effects, and in demonstrating the involvement of protein-interactions taken from generic protein interaction databases. In addition, the authors should more clearly indicate potential biases in their strategy, with regard to genome coverage and overall read-out to explain why classes of expected proteins were not observed (ribosomes). Finally, the actual cellular/biochemical activity of the identified fragments remains unclear: the authors stick to using 'post-transcriptional gene regulation' since the reporter system does not allow to distinguish between translation and effects that impact on RNA stability/turnover, thereby restricting the conclusions that can be drawn with regard to mechanism.

Specific comments:

1. Page 9: in the generated library, only half of all yeast genes are represented, meaning that many relevant or even crucial genes will be missed. It will be important to understand if there is a technical reason for this, e.g. caused by the size selection thereby missing small genes, or because certain genomic regions may be resilient to fragmentation. Bias in the library is suggested by the fact that ribosomal genes are not mentioned in the manuscript, where one would expect these to come up from the screen as prime candidates. Are they not represented in the library, or is there activity not detected by the reporter system? In either case an explanation should be provided.

2. It is clear that a unified reporter system is required to (in this case) model post-transcriptional gene regulation. However, this necessarily implies that compromises need to be made in the design of the reporter construct to reduce biological diversity. In this regard, post-transcriptional regulation can be impacted by many aspects, where e.g. UTR length, sequence and structure vary, which are 'read' in different ways by individual proteins. It is unclear how this was chosen with regard to both 5' and 3' UTR. In addition, it should be explained how hairpins in the 3'UTR, used to tether the bait, may interfere with proper translation.

3. The choice to investigate activity of gene fragments instead of full coding regions has the advantage to explain modularity of protein function, as the authors articulate, however it does not help to understand overall function and activity of the full-length protein. The authors test this to a limited extent by the direct comparison of full-length genes and their respective fragments. Yet, the title of this section ('Full-length proteins display qualitatively similar regulatory activity as truncated fragments') is an optimistic conclusion from the data, where the key is in the word 'qualitatively', indicating that the direction of the regulation is the same for the 'handful' (actually 4) of fragments that were selected for validation. In fact, the data (Fig 3d-g) indicate that occurrence of quantitative differences between full-length and fragments are rather rule than exception (3 out of 4). Moreover, in the case of Yap (Fig 3g) it is questionable if the full-length gene would qualify as an inhibitor. Further experiments show that Sro9 is more potent than any of its fragments (Fig 7b), while Cdc48 is far less active (Fig 7e). From these few tested examples it appears that it is very difficult, if not impossible, to extrapolate fragment-based activity to full-length proteins, and therefore the upshot of this study remains unclear with regard to the regulatory role of full-length proteins as the actual biological actors. To the least, this perspective should be highlighted as a limitation of the study.

4. Page 14: authors write that their data indicate that 'regions adjacent to the RNA-interacting domains typically provide regulatory activity for the RBP'. In fact this is only a hypothesis as one of the explanations of their results. To substantiate this, they should score if active fragments have an adjacent RRM.

5. The authors reason that activity of some fragments may be conferred by interacting proteins, and they therefore cross their data with BioGrid. In yeast, many of the proposed proteins (Fig 4d) interact with a very large number of other proteins, e.g. ~700 for PAB1, or even ~1200 for UBP3 and GIS2 i.e. encompassing 10-20% of the yeast proteome. So the first question is if in this analysis the data were normalized for the number of interactions in biogrid? Second, even if normalized, these high numbers raise doubt whether data as shown in fig 4e reflects specific biological relevance – beyond the fact that both GIS2 and PAB1 are well established as translational regulators for multiple mRNAs.

6. Related to point 5, the authors conclude that the interaction of many fragments with Pab1 'may reflect a general organizational principle of eukaryotic post-transcriptional regulation' (page 24). If this is to say that positive regulators interact with the core translational machinery, this is an intuitive if not established model. Moreover, this is only an indirect outcome of the study by crossing the data with BioGrid, hence it does not look strong as a general conclusion from the study.

7. The authors observe a bimodal activity distribution for Gta1-derived fragments (Fig 5). Why was this pattern not observed in first instance when identifying this gene in the

screen? Furthermore it is unclear if the described bimodal pattern reflects a biological function/mechanism, or if it is an artifact of the system (induced expression, engineered reporter). Also it is unclear if similar examples exist for other genes, or if Gta1 is the exception. In the end, the observations look intriguing but they are not conclusive.

8. Why were so many fragments detected for Ccr4 (Fig 6a)? This would not be expected since (as claimed) the library was generated in an unbiased manner, and since Ccr4 is not exceptionally large.

Reviewer #3:

Remarks to the Author:

In this manuscript, Reynaud and colleagues describe a new genome wide tethered functional assay coupled to fluorescence to decipher the role of different protein fragments in the post transcriptional regulation. They analyze in a very elegant procedure how disordered regions in proteins play a major role in this regulation. I would highly recommend this manuscript to be published in NSMB after the following minor revisions:

Figure 1:

Panel a: Did the authors try any mutated form of the query protein sequence and check the effect on the fluorescence?

Panel b and c, it is clear that the change in target expression is significant, however adding the p-values is recommended

Figure2:

Panel d: How were the reads normalized?

Can the authors show the results of the classical activators and repressors?

Figure3:

Panel b: What would the graphic look like if authors consider the top5 and bottom5 fragments of each category?

If we consider Yap1801 and Sym2, more than 70% of the fragments fall in the far left bin (panel a) how come the effect on translation repression is too low (panel b)? Could authors comment on this.

Same for Sbp1 and Ded1. Although Sbp1 has a major effect on translation activation, Ded1 is 100% present in the far right bin. Did the authors check the expression levels of these proteins?

In the same part of the results, in lane 221 authors state that "in other cases, the full length protein had stronger effect than the isolated fragment" could the authors cite those cases? How were they tested? Could a fragment of Sbp1 containing the 2RRMs and RGG be tested?

In lane 224: Authors should describe the fragment of Sro9 and the missing fragment that could activate the translation

Figure4:

Panel b: Which set was considered for this analysis?

Panel c: (lane 259) It would be very meaningful if authors include a conclusion for this panel. Almost all the negative set is between 0-0.5 and the P value (correct?) is very high. Could the authors comment on this?

Lane 277: what's the enrichment score of the IDR compared to other protein domains? It

would be great if authors can state the percentage of IDR in each part (right, center and left)

Panel e: Is there any evidence of the physical interaction between Pab1 and Gis2? Which domain of the proteins is involved? Is it the same domain shown in panel d?

Figure5:

Panel c: did the authors check the RNA expression level of the fragment and the full length?

Panel d: error bars and p values are missing

Panel e: What's the half life of YFP? Why do authors see YFP expression in the non-induced samples? Can RNA degradation be blocked in this assay and check the bimodality?

Panel e-h: n=2, could the authors get n=3?

Panel J: n is missing, so are the error bars and p values.

Panel K: Does the cellular localization between the gtaD603-767 and Gta1 change?

Figure6:

Panel a-b: is there any known mutation that disrupts the activity of Ccr4 fragment?

Panel d: same as Ccr4, is there any mutation that disrupts the Ded1 activity?

Figure 7:

Panel b: n=2, could the authors get a triplicate of the experiment?

Panel c: what is the color code that is used? N=2 and authors could get n=3.

Panel d: Error bars and p values are missing to show the significance.

Panel f, I and J : n=2, authors could get n=3 and add the error bars and p values or represent the mean of the 3 experiments.

Author Rebuttal to Initial comments

Reviewer #1:

In this manuscript, Reynaud et al use an mRNA tethering assay to survey the ability for sections of yeast proteins to alter post-transcriptional regulation. A similar scheme has been used to assay the impact of ~700 RNA binding proteins (RBPs) on mRNA decay and translation in human cells, and a step forward in the current manuscript is that the authors (1) do not restrict themselves to known RBPs by generating their fragment library in an unbiased fashion, and (2) investigate the impact of different portions of individual proteins. The dataset made by the authors is a hypothesis-generating dataset that will be of interest. The authors also used a clever approach to generate their fragment library that, again, could be useful for the community. However, in my opinion, there are substantial weaknesses in the current study that lead me to have significant reservations:

(1) As a general statement, without the corresponding library of full-length proteins, it is very hard to interpret the impact of different fragments and their biological relevance. For example, in figure 6, it's hard to interpret the effect of different fragments for Ccr4p and Ded1p without showing all the fragments and whole protein.

In the specific cases of Ccr4 and Ded1, both proteins have well-established biological roles consistent with the activities we detect in our screen. Ccr4 is a deadenylase enzyme with a central role in mRNA degradation (reviewed in Collart, WIREs RNA 2016; Passmore & Collier, Nat Rev Mol Cell Biol 2021), and fragments of Ccr4 are repressive. Ded1 is an essential translation initiation factor known to interact with and promote translation of many endogenous mRNAs (Hilliker et al., Mol Cell 2011; Gulay et al., Elife 2020), and we see that tethering of Ded1 likewise increases reporter expression.

We do report data from four screen fragments and the corresponding full-length proteins (Fig. 3d through 3g) showing that in each case the fragment and the full-length protein show qualitatively similar effects, although isolated fragments may be stronger or weaker.

More broadly, tethering of full-length proteins is subject to various drawbacks relative to addressing individual fragments as well as potential benefits. Notably, many full-length proteins would retain RNA-binding domains and thus interact with both endogenous targets and reporter transcripts, potentially creating unphysiological higher-order assemblies. We have designed our analyses to take advantage of the particular benefits of our fragment-based screen, for example by mapping active regions within individual proteins; we also avoid drawing conclusions that could be confounded by its limitations. We would also like to highlight that the strong enrichment of RNA-binding proteins—but not the RNA-binding domains themselves— suggests that we are indeed identifying biologically relevant post-transcriptional regulatory activity.

(2) One challenge in using protein fragments generated by a shotgun method is that, as the authors know, there is a nontrivial potential for the fragments to have altered folding, expression level, localization, etc.—these effects introduce numerous false negatives and positives into the dataset, which make some of the general interpretations about enrichment fraught with alternative explanations.

We agree that false negatives are inevitable in our approach, as in any high-throughput experiment. In our library, some functional proteins will surely be missed by virtue of how the fragments have been constructed—and other kinds of libraries would be subject to their own false negatives. These false negatives don't affect the global conclusions that we draw from our data unless they occur in a biased fashion; we generally don't expect this to happen, although (as discussed below) we were careful to avoid interpreting an "enrichment" of intrinsically disordered proteins because of potential biases in false negatives.

We believe that the enrichment we see for RNA-binding proteins argues that false positives are not driving our overall activity measurements. Indeed, many of the strongest activity scores we see arise from known translation and RNA decay factors, consistent with the idea that our approach identified relevant protein functions. We do agree that the fragments we study may show constitutive activity in

isolation, escaping regulatory context in the larger protein. For this reason, we focus our analysis on identifying active fragments and using them to understand how RNA-binding proteins can modulate translation and degradation.

For example, perhaps unstructured regions are easier to express (or function) than structured domains when the fragments are generated this way, and that causes the enrichments seen by the authors.

We agree that unstructured regions may be easier to capture in our library, or more likely to function in isolation. For this reason, we were careful to avoid discussing or interpreting any “enrichment” of disordered regions. We do believe it is useful to mention that we identify disordered regions with activity, especially in the context of analyzing folded domains with activity (Figure 4C), and so we write:

Our screen identified strong activity in fragments lacking an identifiable, folded domain.

A mention of the above caveat has now been included in the discussion section:

Since unstructured regions may be easier to capture in our library, or more likely to function in isolation, we could not determine whether activity was more likely to occur in folded domains or disordered regions. Nonetheless, both degenerate and specific peptide motifs emerged from our survey, suggesting that both modes of actions play important roles in mRNA regulation.

Similarly, for fragments from nuclear proteins, what are their absolute activity scores as opposed to ones from cytoplasmic proteins?

We agree that this is an interesting question, and didn't see a significant difference in active fragments when comparing proteins with nuclear or cytosolic GO annotations. Many bona fide regulators of translation and cytosolic degradation also enter the nucleus, however. For instance, the poly-(A) binding protein Pab1 and the deadenylase Pop2 (Fig. 1) both localize to the nucleus as well as the cytosol. We did include a nuclear export sequence in our tethering construct, in order to focus on cytosolic regulation; nonetheless, nuclear degradation and nuclear retention are post-transcriptional regulatory mechanisms acting on endogenous genes and could explain some of the regulatory effects we see in our screen.

The lack of control experiments and exploring alternative hypotheses, even for a subset of fragments, make interpretation of the data challenging.

We would certainly like to address alternative hypotheses for measured activity in our screen, besides changes in mRNA translation and decay. We do believe that our RFP control addresses many non-specific and indirect effects, such as changes in cell size or global impacts on protein synthesis. It would

be helpful to hear of the alternative hypotheses under consideration and control experiments that are proposed to address them.

(3) Another major caveat comes in terms of the library size. The authors state “We adapt the tethering assay to survey regulatory activity across the entire yeast proteome in an unbiased manner,” but it is unclear how many genes are represented in the library, the fraction of each gene covered, the coverage for represented amino acids, etc. Similarly, given that the authors use a genomic shotgun approach, it would be useful to discuss false positives in terms of fragments generated from noncoding regions of the genome?

We carried out selection for genomic fragments containing intact reading frames (Figure 2b) with an average size of 479 bp (Figure 2c). This selection largely excludes such intergenic fragments and allows us to focus our analysis on expressed protein fusions. It also excludes highly unstable fragments, and fragments with toxic effects when overexpressed—again allowing us to focus our analysis on fusions that can be meaningfully assessed. The overall library covered 40% of yeast proteins with a median coverage of 41%.

How many of these are represented in the final library? Are any effects observed? A more thorough discussion and analysis of the final library would help substantially.

We now include a more in-depth discussion of the library generation, including tradeoffs and limitations, in the discussion section of the manuscript.

(4) How many replicates of the screen were performed? How reproducible were the scores? As far as I could see, I could not see this described.

We now report a second replicate of our screen that agrees quantitatively with the data set described in our original submission. We present the comparison between the two screens in the Extended Data Fig. 2b. Activity scores in this replicate screen are larger in magnitude and reach the extreme upper or lower limit, which makes it challenging to combine the two data sets. We attribute the rescaled activity scores to differences in FACS gating, and our main analyses still rely on the screen with better resolution of the activity scores across all fragments. Nonetheless, we report a correlation of $r = 0.7$ between these replicate experiments for fragments passing a minimum abundance threshold in both replicates. This value, while strong, underestimates the extent of agreement because of the non-linear saturation of the second replicate screen.

We would highlight that the inclusion of multiple barcodes (on average 3 to 4) for each fragment also provides internal replication for each screen. We also note the strong and quantitative validation data in Fig. 3 as confirmation of the high-throughput functional data from the screen. The screen activity scores

of the validated fragments have a correlation of $r = 0.91$ with direct measurements of the fluorescence ratio by flow cytometry, using independently generated tethering constructs (Fig. 3c).

(5) How many fragments did the authors call as having a high absolute activity score? How many genes were represented? How did this change with different thresholds?

Fragments with an absolute activity score of 1.0 or higher were considered to have “high” activity scores. With this threshold, 1227 fragments were represented, or around 9% of total fragments. If we were more stringent with a cutoff of 1.5, only 403 fragments would be represented.

(6) In general, some of the follow up experiments (e.g., the GTA1 section) feel underdeveloped and do not add much to the manuscript.

We feel the inclusion of the GTA1 data adds biological context to the screening results and highlights that the activity identified in fragments is recapitulated by full-length proteins.

Minor comments:

(1) “Our screen identified strong activity in fragments.” Example data here or a pointer to the results would be useful.

We believe the reviewer is referring to the point that, “Our screen identified strong activity in fragments lacking an identifiable, folded domain.” We present a substantial analysis of two prominent examples in Fig. 6.

Reviewer #2:

In this manuscript, Reynaud et al designed and employed a screening approach to identify proteins and their domains that modulate post-transcriptional regulation of mRNA in yeast. They achieve this by tethering randomly-generated fragments of genomic DNA to a reporter system, consisting of a YFP reporter whose activity was read by FACS sorting. In particular, the increase or decrease of YFP relative to RFP in the control was used to indicate translational activation or repression, respectively. Identity of regulatory fragments was performed by deep sequencing, and aggregate data were investigated to reveal global patterns in proteins and domains that were recovered. In addition, selected candidates were examined, identifying examples where unstructured domains outside RNA-interaction domains contributed to regulation of translation. The authors imply a network of proteins for post-transcriptional gene regulation.

The paper introduces an interesting and potentially powerful approach to identify genes and gene-fragments involved in post-transcriptional gene regulation, by tethering baits of interest (either in a library or in a targeted fashion) to a reporter system. Here, a tagged and size-selected genomic library was generated to span the coding yeast genome. The study convinces in demonstrating that this system can be used to identify known and novel protein domains with regulatory activity in post-

transcriptional gene expression, however it is weaker in other aspects, especially with regard to indicating overall relevance for such activity in full-length proteins, in explaining observed bimodal effects, and in demonstrating the involvement of protein-interactions taken from generic protein interaction databases. In addition, the authors should more clearly indicate potential biases in their strategy, with regard to genome coverage and overall read-out to explain why classes of expected proteins were not observed (ribosomes). Finally, the actual cellular/biochemical activity of the identified fragments remains unclear: the authors stick to using ‘posttranscriptional gene regulation’ since the reporter system does not allow to distinguish between translation and effects that impact on RNA stability/turnover, thereby restricting the conclusions that can be drawn with regard to mechanism.

Specific comments:

1. Page 9: in the generated library, only half of all yeast genes are represented, meaning that many relevant or even crucial genes will be missed. It will be important to understand if there is a technical reason for this, e.g. caused by the size selection thereby missing small genes, or because certain genomic regions may be resilient to fragmentation.

Technical factors, such as those mentioned by the reviewer, likely do account for unequal gene representation in the library. Our selection for in-frame fusions also removes proteins that cannot be expressed in such a context—perhaps because they are toxic, or simply very unstable. Of course, toxic or highly unstable fusions would not be assessed accurately in the functional tethering screen either, and so eliminating these unwanted fusions in advance of the final screen is arguably advantageous. We take care throughout to compare active regulators against the composition of the overall library—rather than the overall proteome—so our conclusions do not reflect technical biases in library composition. We have also added discussion of these limitations of the library in the “Discussion” section.

Bias in the library is suggested by the fact that ribosomal genes are not mentioned in the manuscript, where one would expect these to come up from the screen as prime candidates. Are they not represented in the library, or is there activity not detected by the reporter system? In either case an explanation should be provided.

We actually don’t expect that tethering a single ribosomal protein would increase translation. Nearly all ribosomal proteins are incorporated into the ribosome in the nucleolus, and we wouldn’t expect a free, isolated ribosomal protein to recruit the ribosome. In fact, probably because of the need to balance stoichiometry across dozens of proteins, eukaryotic cells degrade free ribosomal proteins that are present in excess (Sung et al., MBoC 2016).

Consistent with these explanations, we tested 19 fragments from 11 small subunit proteins and saw no statistically significant activity difference relative to our library overall ($p \sim 0.6$, Wilcoxon signed-rank

test). Likewise, we tested 47 fragments from 20 large subunit proteins, again without notable activity ($p \sim 0.9$, Wilcoxon signed-rank test).

However, there are many proteins that associate with the assembled ribosome and compose part of the overall translation machinery. For instance, the translation initiation factor eIF3 is a multi-protein complex that binds the small ribosomal subunit, and contributes some of the strongest activators in our data set. Other “translation machinery associated” or “TMA” proteins showed up in our screen, with activity varying from strong activation (Tma46) to moderate repression (Tma17 and Tma108) of mRNA expression.

2. It is clear that a unified reporter system is required to (in this case) model post-transcriptional gene regulation. However, this necessarily implies that compromises need to be made in the design of the reporter construct to reduce biological diversity. In this regard, posttranscriptional regulation can be impacted by many aspects, where e.g. UTR length, sequence and structure vary, which are ‘read’ in different ways by individual proteins. It is unclear how this was chosen with regard to both 5’ and 3’ UTR. In addition, it should be explained how hairpins in the 3’UTR, used to tether the bait, may interfere with proper translation.

We agree that our study reflects regulatory effects on one model transcript and we now discuss this in the “Discussion” section, along with the potential impact of hairpins.

3. The choice to investigate activity of gene fragments instead of full coding regions has the advantage to explain modularity of protein function, as the authors articulate, however it does not help to understand overall function and activity of the full-length protein. The authors test this to a limited extent by the direct comparison of full-length genes and their respective fragments. Yet, the title of this section (‘Full-length proteins display qualitatively similar regulatory activity as truncated fragments’) is an optimistic conclusion from the data, where the key is in the word ‘qualitatively’, indicating that the direction of the regulation is the same for the ‘handful’ (actually 4) of fragments that were selected for validation. In fact, the data (Fig 3dg) indicate that occurrence of quantitative differences between full-length and fragments are rather rule than exception (3 out of 4). Moreover, in the case of Yap (Fig 3g) it is questionable if the

full-length gene would qualify as an inhibitor. Further experiments show that Sro9 is more potent than any of its fragments (Fig 7b), while Cdc48 is far less active (Fig 7e). **From these few tested examples it appears that it is very difficult, if not impossible, to extrapolate fragmentbased activity to full-length proteins,** and therefore the upshot of this study remains unclear with regard to the regulatory role of full-length proteins as the actual biological actors. To the least, this perspective should be highlighted as a limitation of the study.

Many lines of evidence argue that our fragment-based activity measurements capture functionally relevant information about full-length proteins. Active fragments are strongly enriched for RNA-binding proteins (Fig. 3b), relative to the overall library, and for interactions with core RNA metabolism factors (Fig. 3d). We also did confirm regulatory activity for other full-length proteins, including Gta1 (Fig. 5c) and showed that the fragment we identified in our screen is required for this activity (Fig. 5e,ff). Pab1 is the classic example of an activator in the tethered-function assay (Coller & Wickens, 1998), and we identify 10 activating fragments of Pab1 (activity scores 1.4 – 1.9) out of 11 in our library (Extended Data Fig. 2b) in addition to tethering the full-length protein in our validation experiments (Fig. 1c). We also note that Ccr4 is a deadenylase enzyme with a central role in mRNA degradation (reviewed in Collart, WIREs RNA 2016; Passmore & Coller, Nat Rev Mol Cell Biol 2021), and fragments of Ccr4 are repressive. Ded1 is an essential translation initiation factor known to interact with and promote translation of many endogenous mRNAs (Hilliker et al., Mol Cell 2011; Gulay et al., Elife 2020), and we see that tethering of Ded1 likewise increases reporter expression.

4. Page 14: authors write that their data indicate that ‘regions adjacent to the RNA-interacting domains typically provide regulatory activity for the RBP’. In fact this is only a hypothesis as one of the explanations of their results. To substantiate this, they should score if active fragments have an adjacent RRM.

We agree that “adjacent” is inaccurate and we have changed this to read “outside of RNAinteracting domains”

5. The authors reason that activity of some fragments may be conferred by interacting proteins, and they therefore cross their data with BioGrid. In yeast, many of the proposed proteins (Fig 4d) interact with a very large number of other proteins, e.g. ~700 for PAB1, or even ~1200 for UBP3 and GIS2 i.e. encompassing 10-20% of the yeast proteome. So the first questions is if in this analysis the data were normalized for the number of interactions in biogrid?

The analysis accounted for the large numbers of interactions for Pab1 and other proteins in BioGRID. We used a hypergeometric test for over-representation of activators, versus the overall library, in Pab1 interactors.

Second, even if normalized, these high numbers raise doubt whether data as shown in fig 4e reflects specific biological relevance – beyond the fact that both GIS2 and PAB1 are well established as translational regulators for multiple mRNAs.

These known translational regulators emerged as significant from our enrichment analysis, while many other proteins with hundreds of annotated interactions did not. We report all statistically significant enrichments (Fig. 4d), and did not focus our analysis on posttranscriptional regulators. As such, this

enrichment reflects a biological result showing that active fragments are associated with known regulators of translation relative to the overall library.

6. Related to point 5, the authors conclude that the interaction of many fragments with Pab1 ‘may reflect a general organizational principle of eukaryotic post-transcriptional regulation’ (page 24). If this is to say that positive regulators interact with the core translational machinery, this is an intuitive if not established model. Moreover, this is only an indirect outcome of the study by crossing the data with BioGrid, hence it does not look strong as a general conclusion from the study.

We agree that this model is intuitive and derives from many nice studies of individual proteins. However, we are able to confirm that it holds for proteins chosen from a random library based purely on their regulatory activity. This result is free of ascertainment bias in selecting proteins to study, and thus provides a distinctive contribution.

We believe that integrative analysis of our experimental results with existing data sets and databases is a strength of our work.

7. The authors observe a bimodal activity distribution for Gta1-derived fragments (Fig 5). Why was this pattern not observed in first instance when identifying this gene in the screen? Furthermore it is unclear if the described bimodal pattern reflects a biological function/ mechanism, or if it is an artifact of the system (induced expression, engineered reporter). Also it is unclear if similar examples exist for other genes, or if Gta1 is the exception. In the end, the observations look intriguing but they are not conclusive.

Our screen was designed to measure an average shift in fluorescence intensity, based on sorting cells into four gates (Fig. 2). We saw clear, quantitative agreement between sorting-based activity scores and direct flow cytometry (Fig. 3c), but lacked the resolution to see more complicated changes in the fluorescence distribution, such as the emergence of bimodality. Notably, most other tethered-function experiments rely on bulk measurements that would also obscure bimodality and so this is not a unique limitation of our screen.

We have not seen bimodality in other tethering constructs, out of ~20 other proteins analyzed in this study (e.g., in Fig. 1d and Fig. 3d through 3g). Because we see this only for Gta1, it seems more likely to reflect a distinctive feature of this protein rather than an artifact of the system.

8. Why were so many fragments detected for Ccr4 (Fig 6a)? This would not be expected since (as claimed) the library was generated in an unbiased manner, and since Ccr4 is not exceptionally large.

Our library generation was not biased according to known gene function, but did not provide perfectly even coverage of the proteome for technical reasons. We have added further discussion of these limitations, along with our hypotheses for why certain genomic regions may have been favored over others, to the “Discussion” section. ^[P]_[SEP]

Reviewer #3:

In this manuscript, Reynaud and colleagues describe a new genome wide tethered functional assay coupled to fluorescence to decipher the role of different protein fragments in the post transcriptional regulation. They analyze in a very elegant procedure how disordered regions in proteins play a major role in this regulation. I would highly recommend this manuscript to be published in NSMB after the following minor revisions:

Figure 1:

Panel a: Did the authors try any mutated form of the query protein sequence and check the effect on the fluorescence?

The query protein shown here is part of an overall schematic, designed to help the reader understand how the tethering assay works prior to showing data for actual proteins. All of our results are measured relative to an inactive fusion to the HaloTag protein.

Panel b and c, it is clear that the change in target expression is significant, however adding the p-values is recommended

We now report a multiple regression analysis of regulatory protein, protein-RNA tether, and fluorescent reporter. This analysis is quite robust (adjusted R² = 0.996) and confirms our interpretation; we see significant ($p < 0.001$) effects from Pab1 and Pop2, as well as a significant ($p < 0.001$) interaction between Pop2 and protein-RNA tether, but no other significant terms in the model.

Figure2:

Panel d: How were the reads normalized?

When computing activity scores, reads were normalized by the total number of mapped barcode reads in a library. In some cases (e.g., Fig. 2e and 2f) we report raw read counts.

Can the authors show the results of the classical activators and repressors?

We've now added two panels showing these plots for fragments for the classical activator Pab1 (poly(A)-binding protein) and the classical repressor Cth1 (involved in mRNA degradation). See Extended Data Fig. 2b,c.

Figure3:

Panel b: What would the graphic look like if authors consider the top5 and bottom5 fragments of each category?

The most extreme 5 fragments on each side are >99% sorted into the left-most or right-most bins, similar to Ebs1, Sbp1, and Ded1. The skew for these proteins are very extreme (see plots made for Pab1 and Cth1, Extended Data Fig. 2b,c).

If we consider Yap1801 and Sym2, more than 70% of the fragments fall in the far left bin (panel a) how come the effect on translation repression is too low (panel b)? Could authors comment on this.

We used four equal-width quartile gates (Fig. 2d), and the fluorescence within our population was narrowly distributed (see e.g. Fig. 3d through 3g), so moderate-strength repressors could shift a large fraction of the population into the far-left gate. This effect can be seen for Yap1801 in Fig. 3g, where the Yap1801(374-527) median fluorescence decreases less than 2-fold, but the distribution (light blue) is largely non-overlapping with the control (grey). The activities we infer from sorting and sequencing nonetheless agree quantitatively with measurements of median fluorescence change (Fig. 3c), although they saturate for strong repressors.

Same for Sbp1 and Ded1. Although Sbp1 has a major effect on translation activation, Ded1 is 100% present in the far right bin. Did the authors check the expression levels of these proteins?

The Sbp1(14-178) distribution (Fig. 3d, light pink) has essentially no overlap with the control population (grey). Expression of these proteins was tracked through BFP fluorescence, which is now shown in Extended Data Fig. 3d.

In the same part of the results, in lane 221 authors state that “in other cases, the full length protein had stronger effect than the isolated fragment” could the authors cite those cases?

This is referring to the Sro9 tethering data in Fig. 3e and we have modified the text to include “... such as Sro9”.

How were they tested?

Clonal populations of cells expressing each fragment were analyzed by flow cytometry, using the approach described in Fig. 1.

Could a fragment of Sbp1 containing the 2RRMs and RGG be tested?

Sbp1 is a very small protein containing essentially the two RRM (amino acids 37-119 and 187-270 out of 294) and the intervening RGG repeat (amino acids 125-167), thus the full length version of the protein is essentially the “fragment” in this question.

In lane 224: Authors should describe the fragment of Sro9 and the missing fragment that could activate the translation

In Fig. 7a,b, we show data for these fragments.

Figure4:

Panel b: Which set was considered for this analysis?

We considered any protein that showed up in at least two of the four data sets listed in Fig. 4a. We now describe this in the text:

...we compiled a list of budding yeast RBPs from proteins appearing in at least two of four overlapping datasets that reported RNA-protein interactions

Panel c: (lane 259) It would be very meaningful if authors include a conclusion for this panel. Almost all the negative set is between 0-0.5 and the P value (correct?) is very high. Could the authors comment on this?

The next paragraph, beginning on line 260, provides our interpretation of this panel. For the negative set, the number of Pfam protein families represented includes a small number of genes, thereby resulting in less significant (high) p-values.

Lane 277: what's the enrichment score of the IDR compared to other protein domains? It would be great if authors can state the percentage of IDR in each part (right, center and left)

We are concerned that unstructured regions may be easier to capture in our library, or more likely to function in isolation. For this reason, we feel it is important to be cautious when interpreting “enrichment” of disordered regions. We do believe it is useful to mention that we identify disordered regions with activity given their prominent role in post-transcriptional regulation.

Panel e: Is there any evidence of the physical interaction between Pab1 and Gis2? Which domain of the proteins is involved? Is it the same domain shown in panel d?

Yes; Rojas, M., Wolin, S., et al., (<https://doi.org/10.1371/journal.pone.0052824>) found physical interaction with Pab1 and Gis2 through co-immunoprecipitation experiments. This work was done with

full length versions of each protein, thus it is unknown which domains are interacting in this previous study.

Figure 5:

Panel c: did the authors check the RNA expression level of the fragment and the full length?

We currently provide BFP-expression levels of the Gta1 fragment and full length. These data represent the fluorescent readouts of the BFP protein that was fused to the C-terminal end of the tethered proteins. These data are available in Extended Data Fig. 5a.

Panel d: error bars and p values are missing

We have added a p-value confirming statistical significance of the difference. We chose to show all three replicates rather than include error bars as we felt this was a more informative way to show the data distribution, here and in subsequent sections.

Panel e: What's the half life of YFP? Why do authors see YFP expression in the non-induced samples?

The YFP reporter, and the RFP normalizer, are constitutively expressed. Only the Gta1 tethering construct is induced (see Fig. 1), allowing us to track the change in YFP expression when we switch on Gta1 tethering.

Can RNA degradation be blocked in this assay and check the bimodality?

We don't know of a way to block bulk RNA degradation, especially without greatly perturbing cell physiology. We believe that reduced YFP mRNA levels, accompanied by the decline in YFP expression relative to an RFP control expressed from the same promoter, strongly argues for an mRNA stability effect.

Panel e-h: n=2, could the authors get n=3?

All experiments show very strong reproducibility in replicates with two independently generated clones and very clear differences between tethering constructs. It would be challenging to reconstruct a third independent clone.

Panel J: n is missing, so are the error bars and p values.

We have replaced this panel and included this information for the new data.

Panel K: Does the cellular localization between the gtaD603-767 and Gta1 change?

This is an interesting question, but we were not able to visualize Gta1 localization here.

Figure 6:

Panel a-b: is there any known mutation that disrupts the activity of Ccr4 fragment? Panel d: same as Ccr4, is there any mutation that disrupts the Ded1 activity?

For both proteins, there are clear studies on mutations within the folded domains that impact enzymatic activity. These mutations fall outside of the disordered regions that comprise the most active fragments from our screen.

Figure 7:

Panel b: n=2, could the authors get a triplicate of the experiment? Panel c: what is the color code that is used?

N=2 and authors could get n=3.

Panel d: Error bars and p values are missing to show the significance.

Panel f, I and J : n=2, authors could get n=3 and add the error bars and p values or represent the mean of the 3 experiments.

For flow cytometry experiments in panels (b), (c), (f), and (j), we have confirmed statistical significance in all comparisons (except the ones where we interpret no effect) with the data presented. We think the extent of quantitative agreement between replicates and the differences between constructs are clear from biological duplicates; this can be seen in panels prepared for the revised manuscript, where we are able to plot both replicates for each construct.

In bar plots (d), (e), and (g), all individual measurements are plotted and different shades are used to denote different replicates. We have added this explanation to the figure legend when it is not shown in a figure key.

Decision Letter, first revision:

Message: Dear Dr. Ingolia,

Thank you for submitting your revised manuscript "Surveying the global landscape of

post-transcriptional regulators" (NSMB-A45163A). Please accept my sincere apologies in sending you decision on your study, I am afraid the referees were unavoidably delayed in sending their reports. The reviewers find that the paper has improved in revision, and therefore we'll be happy in principle to publish it in Nature Structural & Molecular Biology, pending minor revisions to comply with our editorial and formatting guidelines.

Sincerely,

Carolina

Carolina Perdigoto, PhD
Chief Editor
Nature Structural & Molecular Biology
orcid.org/0000-0002-5783-7106

Reviewer #2 (Remarks to the Author):

The authors have addressed my concerns appropriately, and I recommend publication of this exciting work.

Reviewer #3 (Remarks to the Author):

I am quite happy about the reply of the authors and suggest to accept the paper for publication.

Final Decision Letter:

Message Dear Dr. Ingolia,

:

We are now happy to accept your revised paper "Surveying the global landscape of post-transcriptional regulators" for publication as a Article in Nature Structural & Molecular Biology. Please accept my sincere apologies for the delay in sending the final decision - I thank you for your patience.

Acceptance is conditional on the manuscript's not being published elsewhere and on there

being no announcement of this work to the newspapers, magazines, radio or television until the publication date in Nature Structural & Molecular Biology.

Your paper will be published online soon after we receive proof corrections and will appear in print in the next available issue. You can find out your date of online publication by contacting the production team shortly after sending your proof corrections. Content is published online weekly on Mondays and Thursdays, and the embargo is set at 16:00 London time (GMT)/11:00 am US Eastern time (EST) on the day of publication. Now is the time to inform your Public Relations or Press Office about your paper, as they might be interested in promoting its publication. This will allow them time to prepare an accurate and satisfactory press release. Include your manuscript tracking number (NSMB-A45163B) and our journal name, which they will need when they contact our press office.

About one week before your paper is published online, we shall be distributing a press release to news organizations worldwide, which may very well include details of your work. We are happy for your institution or funding agency to prepare its own press release, but it must mention the embargo date and Nature Structural & Molecular Biology. If you or your Press Office have any enquiries in the meantime, please contact press@nature.com.

Please note that *Nature Structural & Molecular Biology* is a Transformative Journal (TJ). Authors may publish their research with us through the traditional subscription access route or make their paper immediately open access through payment of an article-processing charge (APC). Authors will not be required to make a final decision about access to their article until it has been accepted. <https://www.springernature.com/gp/open-research/transformative-journals> Find out more about Transformative Journals

Sincerely,

Carolina Perdigoto, PhD
Chief Editor
Nature Structural & Molecular Biology
orcid.org/0000-0002-5783-7106
